# Treatment of Oil Production Data under Fines Migration and Productivity Decline

Grace Loi, Cuong Nguyen , Larissa Chequer, Thomas Russell *, Abbas Zeinijahromi and Pavel Bedrikovetsky

Australian School of Petroleum and Energy Resources, University of Adelaide, Adelaide, SA 5005, Australia
* Correspondence: thomas.l.russell@adelaide.edu.au

**Abstract:** Fines migration is a common cause of permeability and, consequently, injectivity and productivity decline in subterranean reservoirs. Many practitioners implement prevention or remediation strategies to reduce the impact of fines migration on field productivity and injectivity. These efforts rely on careful modelling of the underlying physical processes. Existing works have demonstrated the ability to predict productivity decline by quantifying the extent of particle decline at different fluid velocities. Fluid flows in porous media often involve multiple phases, which has been shown in laboratory experiments to influence the extent of particle detachment. However, no theory has directly accounted for this in a particle detachment model. In this work, a new model for fine particle detachment, expressed through the critical retention function, is presented, explicitly accounting for the immobile fines trapped within the irreducible water phase. The new model utilises the pore size distribution to allow for the prediction of particle detachment at different velocities. Further, an analytical model is presented for fines migration during radial flow into a production well. The model accounts for single-phase production in the presence of irreducible water, which has been shown to affect the extent of fines migration significantly. Combining these two models allows for the revealing of the effects of connate water saturation on well impedance (skin factor growth) under fines migration. It is shown that the higher the connate water saturation, the less the effect of fines migration. The appropriateness of the model for analyzing production well data is verified by the successful matching of 10 field cases. The model presented in this study is an effective tool for predicting the rate of skin growth, its stabilization time and final value, as well as the areal distribution of strained particles, allowing for more intelligent well remediation design. Further, the findings of this study can help for a better understanding of the distribution of fines within porous media and how their detachment might be influenced by pore structure and the presence of a secondary immobile phase.

**Keywords:** fines migration; formation damage; oil production; analytical solution; irreducible water

## 1. Introduction

The detachment of in situ particles during subterranean flows and their consequent capture in the small pore throats of natural porous media is a widely occurring and important process for oil and gas reservoirs. This process, referred to as fines migration, results in a significant decline in rock permeability as well as potentially damaging concentrations of produced fine particles in the surface equipment of production wells.

Fines migration has been shown to be a significant factor in various areas of petroleum, water, and environmental engineering. Field studies have demonstrated the impact of fines migration on both the injectivity [1–3] and productivity [4,5] of wells during the extraction of oil and gas. Other situations in which fines migration has been shown to be important include during fluid leak-off when drilling oil and gas wells [6], during the storage of fresh water in shallow aquifers [7], during $CO_2$ geo-sequestration [8], with methane production from coal beds [9], or as a contributing factor to contaminant transport in shallow water aquifers [10,11].

Informed engineering decisions aimed to prevent or mitigate the negative aspects of fines migration require accurate and reliable mathematical models. In most applications, the goal of these models will be to utilize laboratory or literature-derived parameters to provide forecasts for different design scenarios.

Modelling of fines migration involves the consideration of the processes of fines detachment, transport, and straining. Figure 1 shows a schematic of these processes during two-phase flow in porous media. While the calculations presented in this work will include the effects of an immobile second phase (typically water), it will not consider the more complex case in which two or more phases are mobile. This case will inevitably require the modelling of the fluid-fluid interface and its interaction with attached particles [12,13]. Nonetheless, while the presence of an immobile phase has been shown to greatly affect the effect of fines migration [14], a rigorous treatment of production during fines migration with an immobile phase has not been conducted.

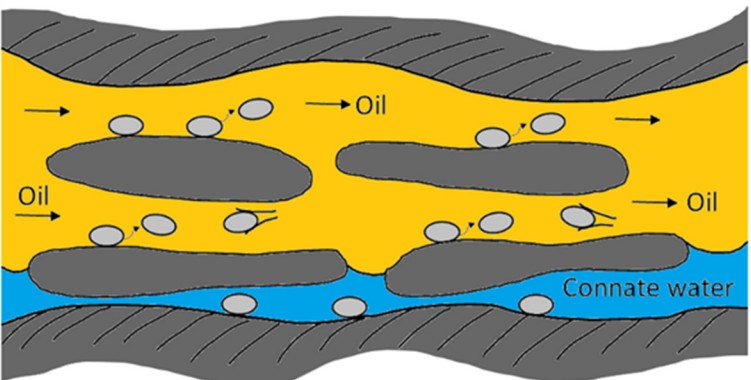

**Figure 1.** Schematic for permeability damage during lifting, migration, and straining of fines in porous space.

The goal of modelling particle detachment is to predict the concentration of detached particles following any change to the system parameters, such as fluid velocity or salinity. To this end, each force acting on attached particles, which depends on the aforementioned system parameters, is calculated, and the balance of these forces is constructed to determine if particle detachment will occur. The most notable forces acting on fines within porous media are the drag, $F_d$, and lift, $F_l$, which arise from flowing fluid, and the electrostatic, $F_e$, and gravitational, $F_g$. For the conditions of interest to this work, the lifting and gravitational forces are several orders of magnitude lower than the electrostatic and drag [15] and so are typically neglected. A schematic of the remaining two forces acting on a spherical particle is presented in Figure 2.

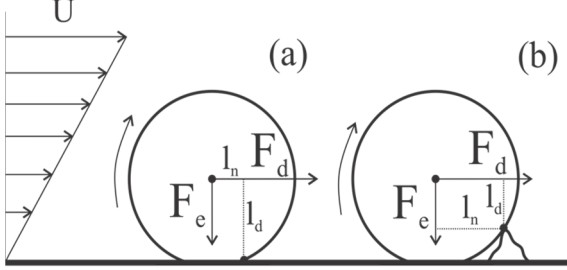

**Figure 2.** Schematic of fines detachment in shear Hele-Shaw flow under the action of capillary, drag, lift, and electrostatic forces: (**a**) lever arms are determined by the contact grain-particle area; (**b**) the lever arms for fine rolling around the rock-surface asperity.

The drag force depends primarily on the size of the attached particle and the fluid velocity within the pore. Higher velocities result in greater drag forces, thus promoting

detachment [16]. This effect explains why particle detachment is more intense in the region around production and injection wells, where the fluid velocity is very high. The electrostatic force depends mainly on the electrostatic properties of the particle, rock, and fluid. Changes to $F_e$ occur primarily by changing the fluid properties, such as during low salinity/smart water injection [17,18] or increasing the temperature [19], which decreases the attaching electrostatic force, promoting particle detachment. The detachment condition takes the form of either a horizontal or vertical force balance or a balance of torques. The two additional parameters, $l_d$, and $l_n$, are the lever arms for the drag and normal (electrostatic) forces, respectively. Once the criterion for detachment has been constructed, it needs to be applied to all attached particles within the porous media to determine the total particle detachment. Various experimental studies have demonstrated that particles detached by changing velocity [20] occur gradually, which is attributed to inhomogeneous particles and pores, i.e., to probabilistic distributions of the coefficients in mechanical equilibrium conditions [21]. Two models have been presented that address this problem. The first works under the assumption that particles form multiple layers on the internal porous surface [22]. When there are multiple layers, fluid flow is restricted to the small volume between particles, and thus the interstitial velocity is very high. This high velocity is more likely to detach particles, resulting in fewer layers of particles. At some layer thickness, an equilibrium is reached where the velocity is insufficient to detach the top layer of particles. The gradual detachment of the layers as the system parameters change to favour detachment is consistent with experimental observations [23]. Importantly, this approach correctly predicts the gradual detachment of particles without accounting for heterogeneity in either the particle or rock properties.

The second approach restricts particle attachment to a monolayer coating of the internal porous surface but allows for a distribution of particle sizes. Larger particles detach more easily due to a strong dependence of the drag force on particle size, and thus it is these particles that get removed during lower velocities or higher salinities. When velocity is increased, or salinity is decreased, smaller and smaller particles are removed from the porous surface. Thus, this approach is also capable of modelling gradual particle detachment.

Both of these approaches fail to account for the distribution of the properties of the porous medium itself. The most important property that varies across the porous space is the pore size, which affects the velocity acting on attached particles. Accounting for varying pore sizes is also critical for describing the distribution of multiple phases within the porous media. Based on the consideration of capillarity, the wetting phase is more likely to occupy smaller pores. Modelling the interaction between the fluid velocity and immobile fluid distributions requires a detachment model that accounts for the pore size distribution.

Particles within the immobile phase experience no drag force and, thus, will not experience detachment. This interaction between particle detachment and multiple fluid phases requires a particle detachment model that accounts for varying pore sizes.

For any manifold of attached particles under strong variation of the fines and rock surface properties, the conditions of mechanical equilibrium determine which particle is detached and which remains attached under given velocity, salinity, pH, temperature, stress, etc. [24,25]. This allows for calculating the attached concentration as a function of the above-mentioned parameters. This dependency is called the maximum retention function (MRF). The maximum retention function for single-phase flow, where fines are detached by drag, depends on velocity. For two-phase flow, when both phases are mobile, the force exerted by the fluid-fluid interface dominates over the drag and attaching electrostatic force [26], resulting in negligible velocity dependence and an MRF that is primarily a function of saturation. In this paper, we discuss an "intermediate" case of oil flow under the presence of connate water. When one phase is immobile, the fluid–fluid interface will not pass over particles, and thus, the capillary force will not affect the MRF. However, it has been shown that the presence of a second phase greatly affects particle detachment even when the second phase is immobile [14]. Without the capillary force and with no

mobile fluid to exert a drag force, particles immersed in the immobile fluid are effectively trapped. As of yet, no MRF model has been presented that accounts for this effect as well as detachment by drag in the mobile phase.

The particular potency of fines migration in reducing rock permeability follows from the ability of the fine particles to clog small pore throats [27]. Despite comprising a relatively low portion of the available porous space, small particles like kaolinite, once suspended in the flow, can easily block pore throats either individually (size exclusion) or by forming larger structures with other particles [28]. Figure 3a shows a scanning electron microscope image of kaolinite booklets coating in an internal porous surface. A schematic representation of the pore-plugging process in Figure 3b shows that a thin large kaolinite platelet can plug even a large pore throat.

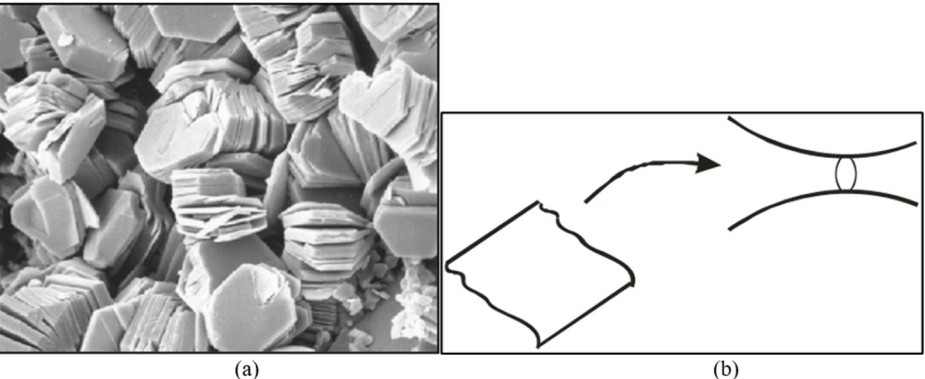

(a)           (b)

**Figure 3.** Scanning electron microscope image of clay particles situated on the rock grains: (**a**) kaolinite booklets consist of the platelets stacked to each other by electrostatic forces; (**b**) kaolinite platelet plugs thin pore throat.

The traditional model for fines migration assumes that the rate of particle capture is directly proportional to the concentration of suspended particles:

$$\frac{\partial \sigma_s}{\partial t} = \lambda c \alpha U \tag{1}$$

where $\sigma_s$ is the concentration of strained (or captured) particles, $t$ is time, $c$ is the suspended particle concentration, $U$ is the particle velocity, $\alpha$ is the particle-fluid velocity ratio, and $\lambda$ is the filtration coefficient.

The direct proportionality with the suspended particle concentration is typically supported by the law of mass action used to arrive at the proportionality between chemical reactions and the concentrations of the reactants. Just as with chemical reactions, we can only be confident in this relationship at low concentrations. At higher concentrations, where particles might interact during the process of capture [28], the capture rate might exhibit a non-linear dependence on the suspended concentration, similar to activity coefficients in chemistry. Other researchers have also found that when there exists a distribution of particle sizes, then the averaging of the system results in a total capture rate with a non-linear dependence on the total suspended particle concentration [29]. In production wells, high particle concentrations in the near-wellbore region mean that any non-linearities in the capture function will be important.

The novelty of this work lies in the derivation of a new particle detachment model that can account for the second immobile phase on the pore scale. As opposed to previous approaches, which produce a smooth critical retention function through particle heterogeneity or particle layering, the new model is built using the pore size distribution of the porous medium. This approach leads to a natural distinction between pores which are filled with the immobile phase, those containing the mobile phase with no detachment, and pores where particle detachment occurs. By connecting the new detachment model with

the well model, the effects of connate water on formation damage due to fines migration can be predicted mathematically for the first time.

The structure of the paper is as follows. Firstly, in Section 2, a new model for particle detachment is presented that accounts for the presence of connate water. Section 3 presents a mathematical model for well productivity that not only allows for any arbitrary particle detachment model but also for arbitrarily non-linear capture functions, $F(c)$. An exact solution for this model is presented in Section 4. In Section 5, a particular particle detachment model is chosen, and the production model is then compared with well data, demonstrating good agreement. Section 5 presents a more general model for particle detachment that allows for any pore size distribution as well as a second immobile phase. Section 6 discusses the results of the work, and conclusions are presented in Section 7.

## 2. Fines Detachment at the Presence of Connate Water

The exact form of the critical retention function can greatly impact the extent of formation damage and growth of skin. In most applications, including those above, a simple form of $\sigma_{cr}(U)$ is used to avoid overly complex parameterisation of the system. While calculations in this study are performed for only a single mobile phase, all oil reservoirs contain an immobile fraction of water. The presence of a second, immobile phase has been shown to reduce the extent of formation damage due to fines migration [14]. To this end, in this section, we derive an expression for a maximum retention function under the presence of connate water and perform its sensitivity with respect to connate water saturation, coefficient of variation of pore sizes, and viscosity ratio.

### 2.1. Formulation of Maximum Retention Function

First, we begin with considering the case where there is no connate water. Suppose the porous media is comprised of a set of parallel cylindrical capillaries with pore radii distributed according to distribution function $f(r_p)$. Spherical particles of constant radius coat the inner layer of these pores and are subject to detachment by the drag force exerted on each particle. While we assume that the coverage of the internal surface by particles is initially spatially homogeneous, we generalise to allow for a partially covered surface, where the function $\gamma(r_p)$ is the fraction of the pore area covered by particles in pores with radius $r_p$.

The flux within each pore is distributed according to their size, resulting in higher drag forces on particles in larger pores (see Appendix C). Thus, for a given velocity, there exists a critical pore radius above which all particles will detach while particles in smaller pores will remain attached. We refer to this critical pore radius as $r_c(U)$. Following the calculations in Appendix B, the critical retention function is given by Equation (2).

When connate water is present in the porous media, some pores will be filled with water and others with oil. As we assume a water-wet rock, the water pores will fill the smallest pores. As we consider only connate water, all particles within the water-filled pores will experience no drag force and hence cannot be detached. Similar to the dry rock case above, we can introduce a critical pore radius, $r_c(S_{wi})$, below which particles are within water-filled pores and thus cannot be detached and above which they are subject to detachment by the oil velocity. These considerations lead to the final form of the critical retention function:

$$\sigma_{cr}(U, S_{wi}) = \frac{8}{3}\phi r_s \frac{\int\limits_{0}^{r_c(S_{wi})} \gamma(r_p) r_p f(r_p) dr_p + \int\limits_{r_c(S_{wi})}^{r_c(U)} \gamma(r_p) r_p f(r_p) dr_p}{\int\limits_{0}^{\infty} r_p^2 f(r_p) dr_p} \tag{2}$$

where the two critical velocities are defined as:

$$r_c(S_{wi}) = r_c|S_{wi} = \frac{V_p^w}{V_p} = \frac{\int_0^{r_c} r_p^2 f(r_p)dr_p}{\int_0^{\infty} r_p^2 f(r_p)dr_p} \tag{3}$$

and

$$r_c(U) = r_p|T(U_p(r_p, U), r_s) = 0 \tag{4}$$

where $T$ is the torque acting on the particle [30]:

$$T(U_p(r_p, U), r_s) = F_e(r_s)l_n(r_s) - F_d(U_p(r_p, U), r_s)l_d(r_s) \tag{5}$$

where $U_p$ is the average velocity within a pore of radius $r_p$, which depends on the total interstitial velocity $U$. Evaluation of this torque balance requires first distributing the flux within each pore and then calculating the corresponding velocity for the drag force acting on the particles. These calculations are presented in Appendix C. The torque balance is used in lieu of the balance of horizontal or vertical forces because it has been shown that these conditions predict particle detachment at higher velocities, and thus, under the conditions of increasing velocity, detachment is predicted entirely by Equation (5).

Being derived from the pore scale, the model (2–5) relies on a number of parameters which can be difficult to identify experimentally. In particular, the pore size distribution often requires micro-computer tomography, which is expensive and not widely available. Using the assumption that the pore size distribution fits a conveniently parameterized distribution function, such as a normal and lognormal distribution, can alleviate this issue. Not only can these distributions be parameterized by only two constants, but the mean pore size can be estimated easily from the rock permeability and porosity. The two critical radii, $r_c(U)$ and $r_c(S_{wi})$, can be obtained using a known pore size distribution and Equations (3) and (4), respectively. The latter only requires the connate water saturation, while the former requires the evaluation of the torque balance. The electrostatic force and lever arms can be evaluated from common laboratory measurements [31].

### 2.2. Sensitivity Analysis

Using the above expressions, the critical retention function can be computed for any velocity and connate water saturation. Figure 4 shows a number of curves with different connate water saturations. Unless otherwise stated, the values used in the calculations are those presented in Table 1. The pore size distribution is modelled using a normal distribution. While the formulation allows for a general function $\gamma(r_p)$ describing the fraction of internal surface area coverage by particles for different pore sizes, for the example calculations below, this function is assumed to be a constant. The initial concentration of attached fines is determined by the conditions of their attachment, i.e., by the velocity in pores of different radii. Therefore, $\gamma$ depends on the pore radius. Since this dependency has not been studied yet, for evaluation purposes in the following sensitivity study, we took an average value.

The curves show that the higher the connate water saturation, the higher the critical retention function. At low velocities, particle detachment occurs in large pores, where $r_p > r_c(U)$. This detachment is uninhibited by the connate water, which inhabits only the smallest pores. As such, all curves coincide for low velocities. At some velocities, the two critical radii will be equal, $r_c(U) = r_c(S_{wi})$. At this point, detachment has occurred in all pores containing oil, and given that detachment does not occur in water-bearing pores, detachment ceases completely. This is the point on the curves where $\sigma_{cr}(U, S_{wi})$ becomes horizontal. The larger the connate water saturation is, the higher the critical retention function is.

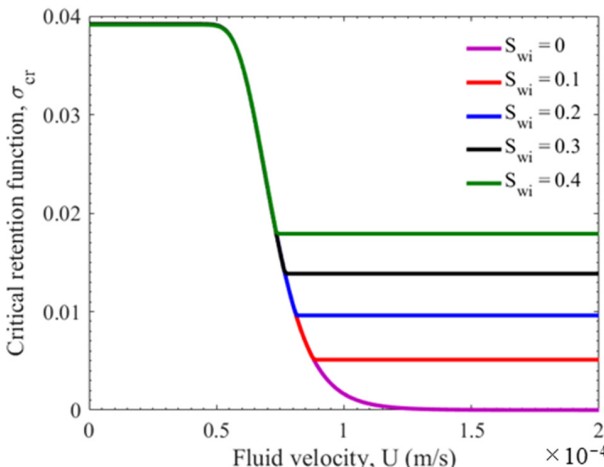

**Figure 4.** Dependence of the critical retention function on the connate water saturation.

**Table 1.** Parameters used for calculations of $\sigma_{cr}(U, S_{wi})$.

| Parameters | Value | Unit |
|---|---|---|
| $r_s$ | $1 \times 10^{-6}$ | m |
| $l_d$ | $2 \times 10^{-6}$ | m |
| $\phi$ | 0.25 | - |
| $\mu = \mu_o/\mu_w$ | 50 | - |
| $\gamma(r_p)$ | 0.3 | - |
| $M_{rp}(m)$ | $5 \times 10^{-6}$ | m |
| $C_v$ | 0.15 | - |
| $\omega$ | 1.7 | - |

Where $M_{rp}$ and $C_v$ are the mean and coefficient of variation of the pore size distribution, and $\omega$ is the drag coefficient used to calculate the drag force acting on particles.

Figure 5a,b show the sensitivity of the curves with the coefficient of variation of the pore size distribution ($C_v$) and the ratio of oil to water viscosity ($\mu$).

Widening the pore size distribution (increasing $C_v$) results in a decrease in the maximum value of the critical retention function, an increase in the minimum value, and, consequently, a decrease in the total detachable fines concentration. The decrease in the total number of fines follows from the fact that the internal surface grows slower with $C_v$ than the internal pore volume does. Mathematically, Equation (2) for the critical retention function can be shown in the limit of $U \to 0$ to be proportional to $1/(1 + C_v)$. The curves also show that when the pore size distribution is wide, detachment occurs over a wider range of velocities as a result of the widening of the distribution of velocities acting on each particle.

The Figure 5b shows the sensitivity with the oil-water viscosity ratio. Increasing the oil viscosity result in a higher detaching drag force, resulting in greater fines detachment. Thus, the critical retention curves are strictly smaller for larger values of $\mu$. The viscosity ratio does not affect either the maximum or minimum values of the critical retention function.

Figure 6 presents the sensitivity of the critical retention function with respect to the particle radius and the fraction of the internal surface area covered by particles, $\gamma$.

Increasing the particle radius increases the critical retention function proportionally, as seen explicitly in Equation (2). For a constant internal area coverage, the number of particles decreases according to $r_s^2$, while the volume of each particle increases proportionally to $r_s^3$. Thus, the total volume increases proportionally to $r_s$. Increasing the fraction of the internal surface covered by particles, $\gamma$, results in a proportional increase in the critical retention function as per Equation (2). More complex dependencies may arise when $\gamma(r_p)$ is not constant.

For these calculations, a normal distribution is used. However, many porous media can exhibit more complex distributions that would consequently result in more complex critical retention functions.

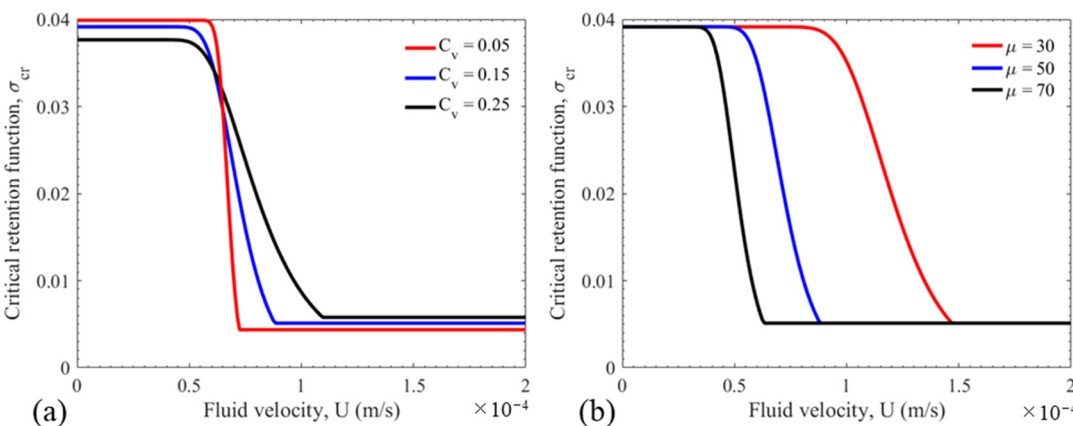

**Figure 5.** Sensitivity of the critical retention function with respect to the coefficient of variation of the pore size distribution (**a**) and the ratio of oil to water viscosity (**b**).

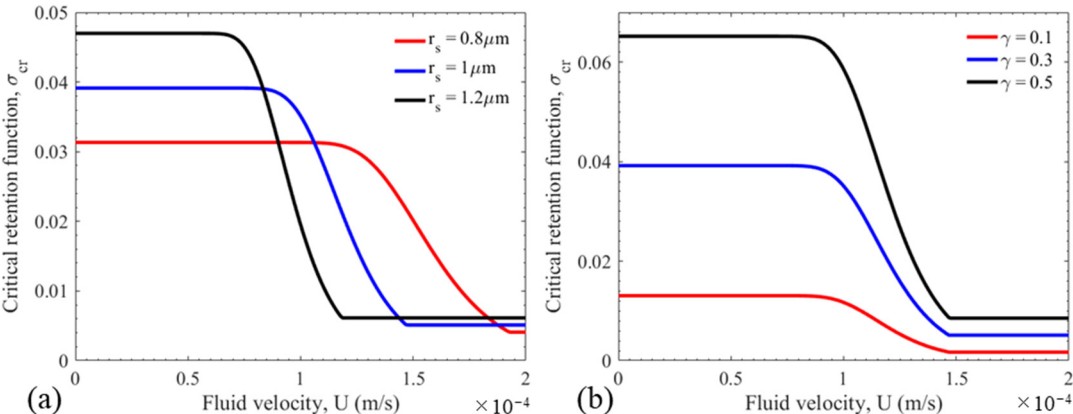

**Figure 6.** Sensitivity of the critical retention function with respect to the particle radius (**a**) and the fraction of internal surface coverage (**b**).

## 3. Mathematical Model for Well Inflow Performance under Fines Migration

This section presents the model assumptions and governing equations for oil flow with fines migration towards the well boundary and initial conditions that reflect fines mobilisation and formulates a one-dimensional (1D) axi-symmetric flow problem in dimensionless coordinates.

### 3.1. Assumptions of the Model

The main assumptions of the model are as follows:

- The incompressibility of the particle suspension and of retained particles;
- The validity of Darcy's law for the flow of oil in the presence of connate water;
- A universal relationship relating the decline in permeability to the concentration of retained particles;
- A linear expression for the kinetics of suspended particle capture, analogous to the active mass law of pores and particles;
- Amagat's law of volume additivity during particle suspension and capture, leading to the flux conservation during particle suspension and retention;
- Instant particle detachment;
- The flow rate is non-increasing with time, following the decrease in rock permeability due to fines migration;
- The well is not hydraulically fractured.

For the flow towards the well with some rate $q$, the velocity can be found from the condition of colloid-suspension incompressibility.

The problem considered here involves the flow of oil towards a well with a constant flow rate, $q$. Under the condition of the incompressibility of the fluid suspension, the fluid velocity can be expressed as follows:

$$U = \frac{q}{2\pi r} \tag{6}$$

where $U$ is the fluid velocity, $q$ is the flow rate per unit of sand thickness, and $r$ is the radial distance from the centre of the wellbore.

During the production of oil, most wells will produce more than one fluid. The detachment and deposition of fines occur during the early stages of production, during which many wells have very low water cuts, allowing us to justify the assumption of single-phase flow. All field cases discussed in this paper describe the single-phase production of oil.

A pressure-diffusivity wave in low-compressibility oil propagates significantly faster than a particle during Darcy's flow towards a well. This ratio is equal to the transient time of establishing a quasi-steady state divided by the exploitation period. This parameter has ab order of magnitude of $10^{-2}$–$10^{-4}$ [32] and is used as a small parameter by [33,34]. Therefore, the assumption of incompressibility is common in reservoir behaviour and well productivity and is not appropriate in well testing. For example, the definition of well productivity and Craft-Hawkins formula for skin assume fluid incompressibility.

### 3.2. Governing Equations for Axi-Symmetric Flow of Suspension-Colloidal Fluid

The mass balance for suspended, attached, and strained particles is

$$r\frac{\partial}{\partial t}[\phi(1 - s_{wi})c + \sigma_a + \sigma_s] - \frac{\partial}{\partial r}(rc\alpha U) = 0 \tag{7}$$

where $t$ is time, $c$, $\sigma_a$, and $\sigma_s$ are the suspended, attached, and strained concentrations, respectively; $\phi$ is the porosity; $s_{wi}$ is the irreducible (or connate) water saturation; $\alpha$ is the drift delay factor. It is assumed here that the average particle velocity is a fixed ratio of the average fluid velocity, such that the average particle velocity is equal to $\alpha U$ with $\alpha \ll 1$. [35].

For a particle suspension with distributed particle sizes, the kinetics of particle capture is as follows [29]:

$$\frac{\partial \sigma_s}{\partial t} = \alpha \lambda F(c)|U| \tag{8}$$

where $F(c)$ is the suspension function, reflecting the heterogeneity in particle capture for different-sized particles. For mono-sized particles,

$$F(c) = c \tag{9}$$

The condition for constant $\lambda$ is satisfied only in the condition where the deposit concentration is small relative to the number of filtration sites in the porous media. When particles are injected into the media, even at small concentrations, this condition is likely to be violated, given sufficient injection time. In this case, the filtration coefficient should be formulated as a function of the deposit concentration [36]. For fines migration, however, the supply of particles is finite and limited to the initial attached concentration. Thus for many fines migration applications, the deposit concentration is small enough to justify constant $\lambda$.

Following Pang and Sharma [37] and Mojarad and Settari [38], the permeability is assumed to be reciprocal to a linear function of the retained particle concentration ($\sigma_s$) where $\beta$ is the so-called formation damage coefficient. Darcy's law accounting for permeability reduction due to particle straining is:

$$U = -\frac{kk_{rowi}}{\mu(1 + \beta\sigma_s)}\frac{\partial p}{\partial r} \tag{10}$$

### 3.3. Initial and Boundary Conditions

It is assumed that the moment production commences, the steady-state velocity distribution throughout the reservoir is established instantaneously. In natural reservoirs, rock and fluid compressibilities result in the propagation of pressure waves into the reservoir, leading to time-dependent velocity and pressure distributions.

The initial conditions for the attached particles are given by the maximum retention function. In the region closest to the wellbore, the velocity can exceed the maximum velocity, $U_m$, resulting in the detachment of all particles. At some point further from the well, the velocity is lower than the critical velocity, $U_i$, and as such, at this point and at all distances further from the well, no particle detachment occurs. In between these two extremes, the extent of particle detachment is determined by the form of the critical retention function. Given the assumption of instantaneous particle detachment, all detached particles will immediately become suspended, and thus, the initial condition for the suspended concentration is given by the difference between the initial attached concentration, $\sigma_{aI}$ and the remaining attached concentration (differing by a factor of the rock porosity):

$$t = 0 : c(r,0) = \begin{cases} \sigma_{aI}\phi^{-1}, & r_w < r < r_m \\ \dfrac{\sigma_{aI} - \sigma_{cr}(U)}{\phi}, \ U = \dfrac{q}{2\pi r} & r_m < r < r_i \\ 0, & r_i < r < r_e \end{cases} \quad (11)$$

where the size of the fines mobilisation zone corresponds to the critical velocity $U_i$

$$r_i = \frac{q}{2\pi U_i} \quad (12)$$

and the size of the zone where all attached particles are mobilised is determined by the maximum velocity $U_m$

$$r_m = \frac{q}{2\pi U_m} \quad (13)$$

The reservoir pressure is assumed to be fixed at some fixed distance from the wellbore, called the drainage radius, $r_e$.

$$r = r_e : c = 0, \ p = p_{\text{res}} \quad (14)$$

Dimensionless equations.

We introduce the following dimensionless variables:

$$X = \left(\frac{r}{r_e}\right)^2, T = \frac{1}{\pi r_e^2 \phi}\int_0^t q(T)dT, \Lambda = \lambda r_e, S_a = \frac{\sigma_a}{\sigma_{aI}}, S_{cr} = \frac{\sigma_{cr}}{\sigma_{aI}}, S_s = \frac{\sigma_s}{\sigma_{aI}}, C = \frac{c\phi}{\sigma_{aI}}, P = \frac{2\pi k k_{rowi}p}{\mu q} \quad (15)$$

Substituting these variables, the system of governing Equations (6)–(10) becomes

$$U = \frac{q}{2\pi r_e\sqrt{X}} \quad (16)$$

$$\frac{\partial(1 - s_{wi})C}{\partial T} - \alpha\frac{\partial C}{\partial X} = -\frac{\partial S_s}{\partial T} \quad (17)$$

$$\frac{\partial S_s}{\partial T} = \alpha\Lambda\frac{F(C)}{2\sqrt{X}} \quad (18)$$

$$\frac{2X}{1 + \beta\sigma_{aI}S_s}\frac{\partial P}{\partial X} = 1 \quad (19)$$

The dimensionless initial and boundary conditions are

$$T = 0 : C(r,0) = \begin{cases} 1, & X_w < X < X_m \\ 1 - S_{cr}\left(\dfrac{q}{2\pi r}\right), & X_m < X < X_i \; ; \quad S_s(r,0) = 0 \\ 0, & X > X_i \end{cases} \tag{20}$$

$$\begin{aligned} X = X_e : P = P_{\text{res}} \\ X = X_i : C(X_i, T) = 0, \; P = P_i \end{aligned} \tag{21}$$

where

$$X_i = \left(\frac{r_i}{r_e}\right)^2 ; \; X_m = \left(\frac{r_m}{r_e}\right)^2 \tag{22}$$

## 4. Analytical Model for Well Inflow Performance under Fines Migration

In this section, we derive the exact solution for oil production with fines migration for both linear and non-linear suspension functions and calculate the dynamics of the skin factor from the analytical model.

Exact solution for non-linear suspension function $F(C)$—the case of high suspension concentrations.

Substitution of Equation (18) into Equation (17) yields

$$\frac{\partial (1 - s_{wi})C}{\partial T} - \alpha \frac{\partial C}{\partial X} = -\alpha \Lambda \frac{F(C)}{2\sqrt{X}} \tag{23}$$

Subject to the initial conditions (20), the non-linear first-order partial differential Equation (23) can be solved by the method of characteristics [39]. The characteristics are straight lines given by

$$X = X_0 - \frac{\alpha}{1 - s_{wi}} T \tag{24}$$

Along these characteristics, Equation (23) transforms into an ordinary differential equation with initial conditions given at the point $(X_0, 0)$:

$$\frac{dC}{dT} = -\frac{\alpha \Lambda F(C)}{2\sqrt{X_0 - \alpha T}} \tag{25}$$

Separating variables in ODE (25) and integrating along the characteristics, we obtain

$$\int\limits_{C(X_0,0)}^{C} \frac{dC}{F(C)} = -\int\limits_{0}^{T} \frac{\alpha \Lambda \, dT}{2\sqrt{X_0 - \alpha T}} \tag{26}$$

The integral on the right-hand side can be expressed exactly, resulting in an implicit solution:

$$\int\limits_{C(X_0,0)}^{C} \frac{dC}{F(C)} = \sqrt{X_0 - \frac{\alpha}{1 - s_{wi}} T} - \sqrt{X_0} \tag{27}$$

The variable $X_0$ distinguishes the different characteristic lines. Expressing it by Equation (24) and substituting it into Equation (27) results in the final solution for the suspended concentration:

$$\int\limits_{C(X + \frac{\alpha}{1 - s_{wi}} T, 0)}^{C(X,T)} \frac{dC}{F(C)} = \sqrt{X} - \sqrt{X + \frac{\alpha}{1 - s_{wi}} T} \tag{28}$$

The initial conditions (20) are divided in the flow domain $X_w < X < 1$ into three regions: $[X_w, X_m]$, $[X_m, X_i]$, $[X_i, 1]$, where the expressions for the initial concentrations are different. This leads naturally to a division of the entire $(X, T)$ space. Figure 7 shows the region boundaries along the characteristics, which begin at points $(X_m, 0)$ and $(X_i, 0)$.

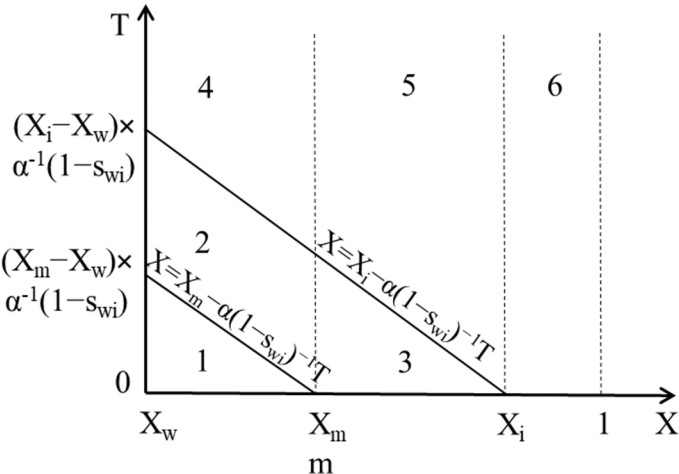

**Figure 7.** Schematic for flow towards well under fines migration: six typical flow zones with different expressions for suspended and retained concentration in the analytical model; $X$ versus $T$ express straight-line trajectories of particle transport, where $X$ is the square of the dimensionless radius, and $T$ is the dimensionless time in PVI.

The initial condition for Equation (28) in Zone 1 is uniform and corresponds to the mobilization of all particles near the wellbore due to the high velocities. Along each of the characteristics within this region, the solution (28) reflects the decrease in the suspended particle concentration as a result of particle capture.

For Zone 3, the initial condition for Equation (28) is given by Equation (20), which describes a non-uniform initial suspended concentration. The non-uniform initial profile travels towards the well along the characteristic lines, and the suspended concentration decreases due to capture. All points in Zone 2 correspond to characteristics with the initial condition in Zone 3.

For regions with all characteristics within Zones 4, 5, and 6, the initial conditions correspond to point $X > X_i$, which corresponds to no detachment. As such, the suspended concentration in these Regions is zero.

For a known suspended concentration $C(X,T)$, the strained particle concentration $S_s(X,T)$ can be calculated by integrating the particle kinetics Equation (18) over time. Given that the expressions for $C(X,T)$ differ for the different Zones, this integration is done sequentially. For example, the strained concentration in Zone 1 is obtained by integrating Equation (18) with zero initial condition. Obtaining $S_s$ in Zone 2 involves integrating Equation (18) in time from the point on the dividing characteristic (starting at $(X_m,0)$) using the value from Zone 1 as the initial condition. The same is repeated for Zone 4, with initial conditions provided on the characteristic starting at $(X_i,0)$; however, this integration is trivial as $C(X,T)$ is zero in Zone 4. The same is true in Zone 5, namely that it inherits the strained concentration from Region 3, at which point it becomes steady-state. The strained concentration in Zone 6 is zero, i.e., there is no formation damage.

For the particular case where the suspension function $F(C)$ is parabolic, Equation (28) can be transformed into an explicit expression for the suspended concentration, despite cumbersome calculations [40].

### 4.1. The Case of Linear Suspension Function F(C) = C

In the case of low suspension concentrations, the solution (28) allows for explicit expressions of $C(X,T)$ and $S_s(X,T)$ in Zones 1–6 for $F(C) = C$; see Appendix A for the derivation.

### 4.2. Calculation of Well Index, Impedance and Skin Factor

Following the assumptions of the model discussed earlier, the following calculations are valid only in the case of quasi-steady state production of non-fractured wells.

The equation for the well productivity index is

$$PI(T) = \frac{q(T)}{\Delta p(T)} \tag{29}$$

which decreases as permeability decreases (formation damage accumulates). The well impedance is the normalised reciprocal of the well productivity index:

$$J(T) = \frac{PI(0)}{PI(T)} \tag{30}$$

The expression for the initial pressure drop follows from the Equation for well performance

$$\Delta p_0 = -\frac{q_0 \mu}{2\pi k k_{rowi}} \ln\left(\frac{r_w}{r_e}\right) \tag{31}$$

The dimensionless initial pressure drop is

$$\Delta P_0 = -\frac{1}{2}\ln(X_w) \tag{32}$$

Adding the pressure drop due to skin $S$

$$\Delta p(t) = -\frac{q_0 \mu}{2\pi k k_{rowi}}\left[\ln\left(\frac{r_w}{r_e}\right) + S(t)\right] \tag{33}$$

Substituting (33) into (29) and (30) results in the relationship between impedance and skin factor

$$J(T) = 1 - \frac{S(T)}{\ln(X_w)} \tag{34}$$

For constant-rate production, the impedance is equal to the normalised pressure drop, which is calculated from Darcy's law (19)

$$J(T) = \frac{\Delta P(T)}{\Delta P_0} = 1 - \frac{\beta \phi S_{aI}}{\ln(X_w)} \int_{X_w}^{1} \frac{S_s(X, T)}{X} dX \tag{35}$$

From Equations (34) and (35), what follows is the expression for skin growth $S(T)$

$$S(T) = \beta \phi S_{aI} \int_{X_w}^{1} \frac{S_s(X, T)}{X} dX \tag{36}$$

Equations (35) and (36) are based on explicit Formulae (A1) to (A17) that can be used for predicting well behaviour as a result of fines migration.

## 5. Analysis of Productivity Decline

The previously described model, whose analytical solution is presented in Equations (A1)–(A17), allows for calculating well productivity decline subject to an explicit expression for the critical retention function.

### 5.1. Effects of Connate Water on Well Productivity

Equipped with the more rigorously defined critical retention function, the well productivity decline can be calculated using Equations (A1)–(A17). This allows for the examination of the effect of connate water directly on the skin growth of the well. Figure 8 provides calculations for three different connate water saturations. Calculations are done with $(\beta, \lambda, \alpha) = (490, 50, 10^{-2})$.

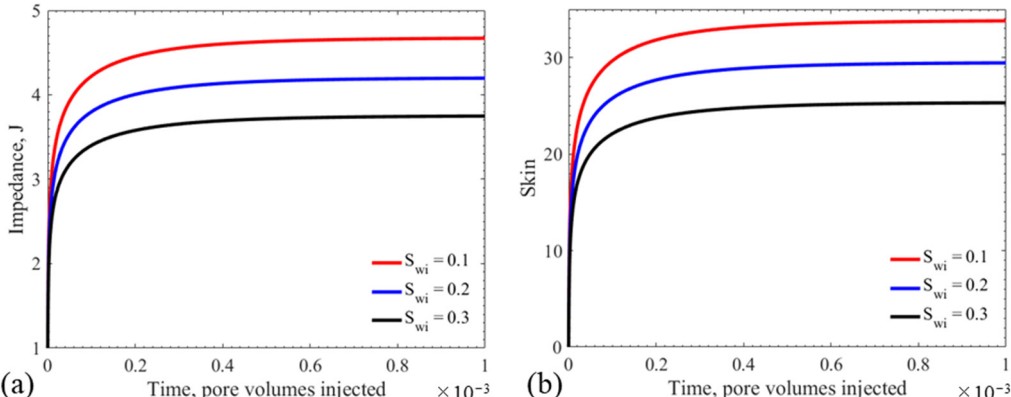

**Figure 8.** Impedance (**a**) and skin growth (**b**) with time (measured in produced volume as a fraction of the reservoir pore volume) for different connate water saturations.

The figures show that with higher values of the connate water saturation, the well experiences less formation damage. This is consistent with the previous observations that the presence of connate water prevents a certain fraction of attached particles from being detached due to drag forces exerted by the oleic phase. This analysis is extended in Figure 9, where the final stabilised value of both the impedance and skin are plotted against the connate water saturation.

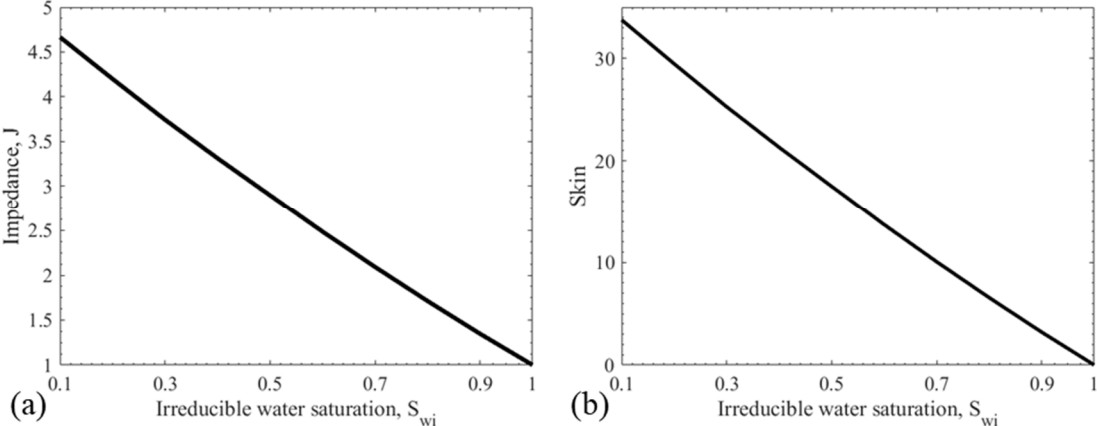

**Figure 9.** Stabilised value of impedance (**a**) and skin (**b**) versus the connate water saturation.

This figure confirms the general tendency for less formation damage in reservoirs with higher connate water saturations. The curves are slightly nonlinear, indicating a greater importance of connate water when its saturation is lower. At $S_{wi} = 1$, all particles are immersed in the immobile connate water, and no formation damage is present. While not shown in these graphs, the lower limit, at $S_{wi} = 0$, corresponds to the dry-rock situation discussed earlier.

### 5.2. Simplified Maximum Retention Function

For the purposes of analysing well data, we introduce a form of the critical retention function that has been shown to broadly capture the behaviour of fines detachment. This form has been demonstrated to match well with experimentally derived MRFs [22,41].

$$\sigma_{cr}(U) = \begin{cases} \sigma_0\left(1 - \left(\frac{U}{U_m}\right)^2\right), & U < U_m \\ 0, & U > U_m \end{cases} \tag{37}$$

where $\sigma_0$ is the maximum retained concentration in the limit of zero velocity, and $U_m$ is the maximum velocity beyond which all particles have been detached.

The MRF given by Equation (37) is two-parametric and provides a simple, explicit relationship between the attached particle concentration and the fluid velocity. The forms of MRF presented in Figures 4–6 and calculated from the microscale model for fines detachment (2) require significantly more constants. To match the well index curve, which is usually the only information from the well productivity history, we chose Equation (37), which contains two parameters alone. However, the form (37) reflects the following features of the curves presented in Figures 4–6: zero derivative at $U = 0$ corresponding to low detachment at small velocities, a monotonically decreasing trend, and no fines detachment beyond some maximum velocity.

The attached particle concentration prior to flow is equal to $\sigma_{aI}$. Thus, detachment does not occur until the velocity is large enough to result in a critical retention function lower than the initial attached concentration. At the point when detachment begins, we define the velocity as the critical velocity:

$$U_i = U | \sigma_{cr}(U) = \sigma_{aI} \tag{38}$$

For velocities smaller than $U_i$, particle detachment does not occur. As per Equation (37), this formulation results in three regions around the wellbore: close to the wellbore, velocity is high such that $U > U_m$ and all particles detach; further from the wellbore, $U_m > U > U_i$, and, thus, detachment is given by Equation (37), and even further from the wellbore, $U_i > U$ and, thus, the velocity is too small to detach any particles.

This formulation of the critical retention function can be simply characterised either by the parameters $(\sigma_0, U_m, \sigma_{aI})$ or equivalently $(U_i, U_m, \sigma_{aI})$. Further in the text, we use the latter.

As opposed to the difficult and expensive calculations required to characterise the critical retention function presented in Section 2, the parameters in Equations (37) and (38) can be determined straightforwardly from coreflooding experiments. During a stepwise increase in injection velocity, treating particle breakthrough concentration and pressure drop across the core with a mathematical model for fines migration allows for determining the total detached concentration. The collection of these values across a wide range of velocities allows the construction of the critical retention function [41]. The initial velocity, $U_i$, can be determined from the first velocity at which particle detachment occurs.

### 5.3. Sensitivity Study

In this section, we investigate the relative importance of the different model parameters on skin growth during fines migration.

Following Equation (36) and the explicit expressions for the strained concentration presented in Appendix A, the skin factor is dependent on the drift-delay factor $\alpha$, filtration coefficient $\lambda$, formation damage coefficient $\beta$, initial concentration of movable fines in the rock $\sigma_{aI}$, maximum velocity $U_m$, and critical velocity $U_i$. Figure 10 shows how these parameters affect skin growth and breakthrough concentration over time. The breakthrough concentration is defined as the cumulative concentration of fines produced at the wellbore, $r = r_w$.

The drift delay factor, $\alpha$, describes the relative velocity of the particles compared to the carrier fluid. Higher values of $\alpha$ signify suspended particles that move faster through the porous medium and thus spend less time between detachment and capture. Correspondingly, Figure 10a shows that the skin growth is faster and reaches the stabilised value sooner the higher the drift delay factor is. The drift delay factor has no effect on the stabilised value of the skin. A similar effect is observed in the breakthrough concentration; higher $\alpha$ results in faster stabilisation but has no effect on the final cumulative produced fines concentration.

The filtration coefficient, also varied in Figure 10a, has a less pronounced effect on the stabilisation time. For higher values of $\lambda$, detached particles travel a smaller distance on average before capture, and thus, the time between detachment and stabilisation (when all particles are captured) is lower. The filtration coefficient also affects the stabilised skin, with

higher values of $\lambda$ corresponding to higher values of skin. While the filtration coefficient does not affect the total concentration of detached particles, it does increase the fraction of detached particles which become strained rather than being produced at $r = r_w$. This is observed in the BTC curve, which shows the largest production of fines for the case with the lowest filtration coefficient.

Figure 10b shows the effect of the formation damage coefficient, $\beta$, on the skin growth of the well. The breakthrough concentration is not included as $\beta$ does not affect the quantity of produced fines. The higher the formation damage coefficient is, the greater the permeability decline induced by each particle and, thus, the larger the skin growth. Figure 10c shows the effect of the initial attached concentration, $\sigma_{aI}$. The results for the skin growth are similar. In fact, in the case where $\sigma_{aI} = \sigma_0$, $\beta$ and $\sigma_{aI}$ appear in the solution for the pressure drop and skin only as the product $\beta\sigma_{aI}$ and thus, their effects are exactly equal. Differing from the formation damage coefficient, however, the initial attached fines concentration also increases the breakthrough concentration of fine particles.

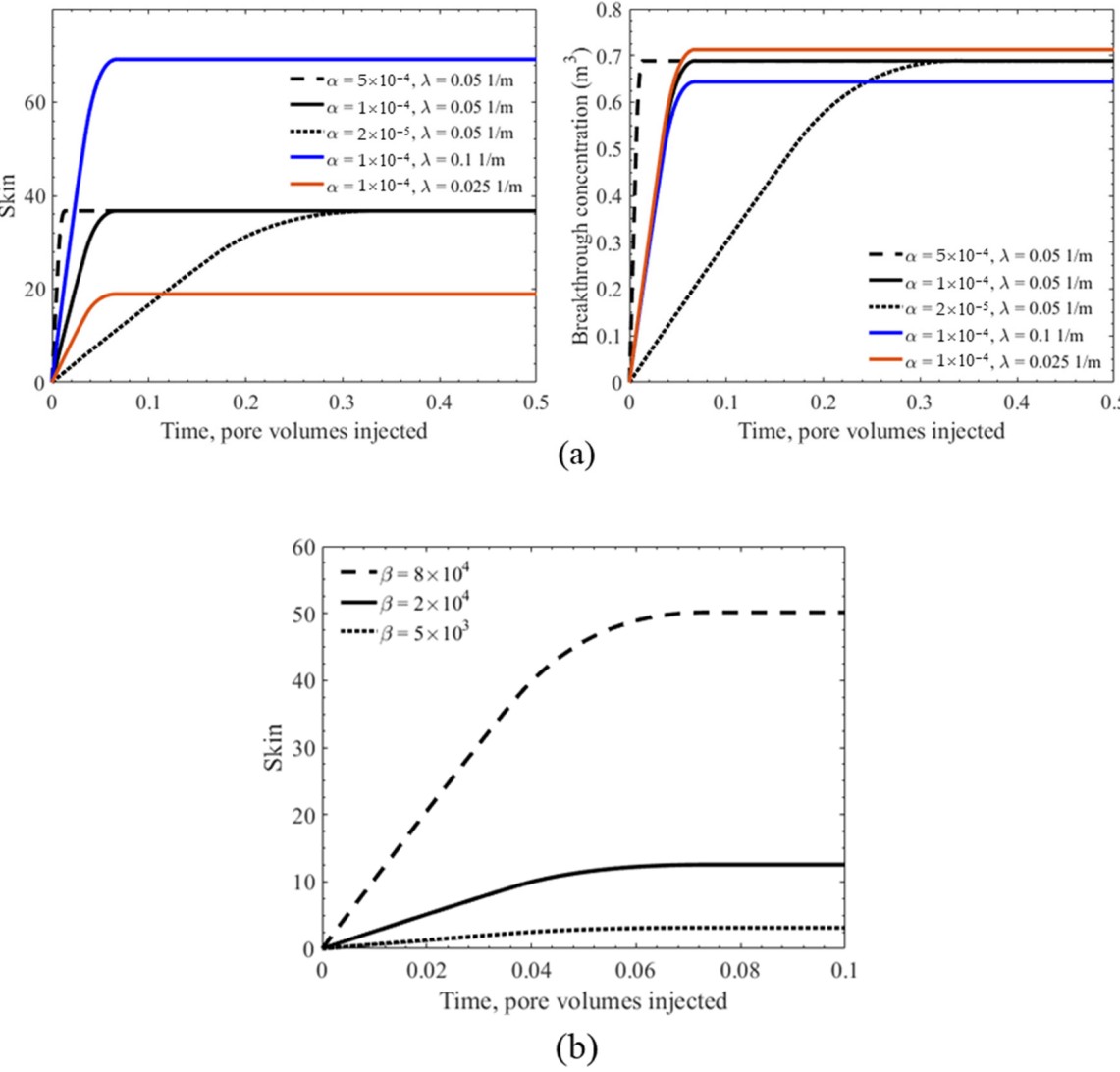

**Figure 10.** *Cont.*

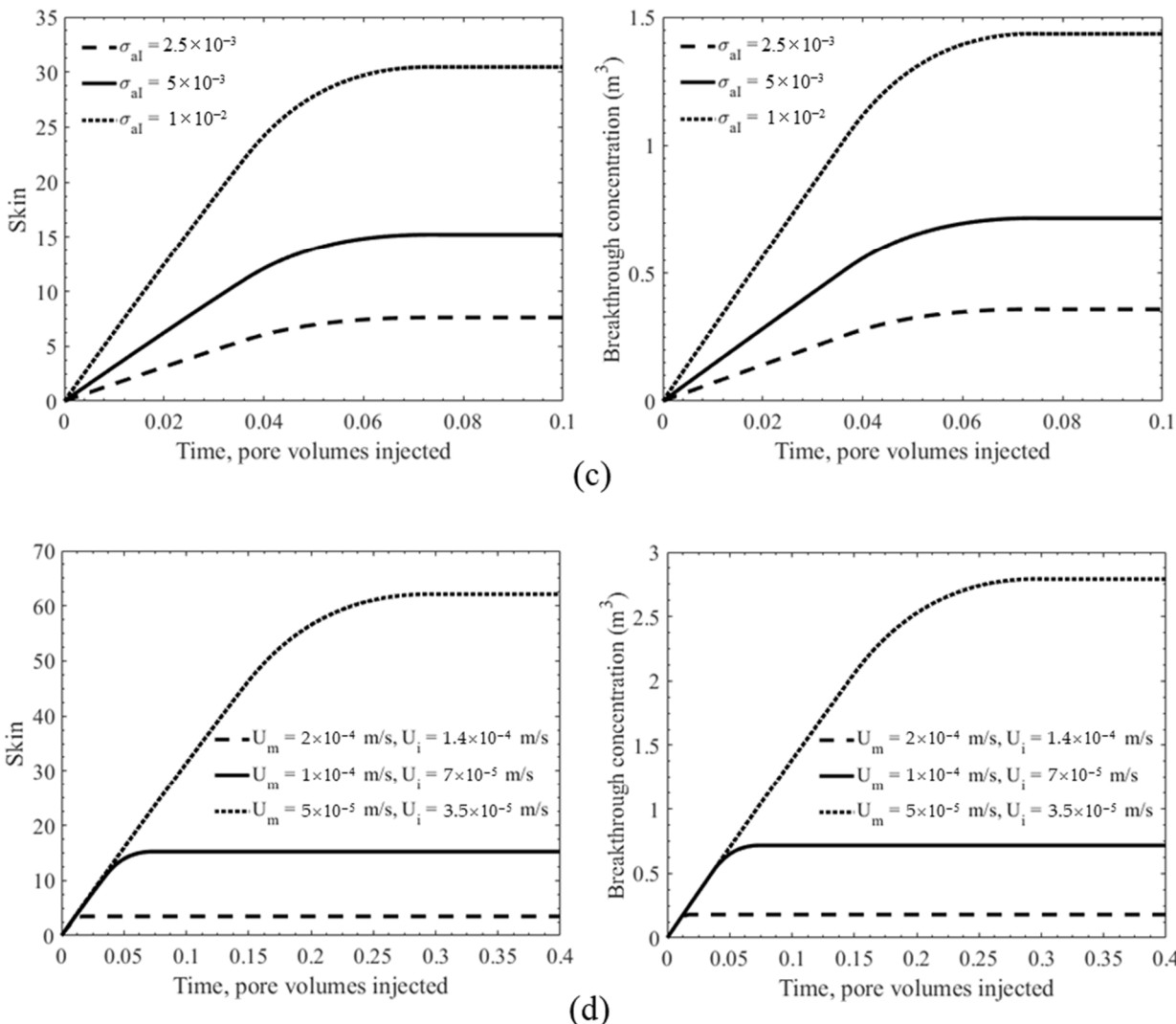

**Figure 10.** Sensitivity analysis for fines-migration skin factor and breakthrough concentration with respect to (**a**) drift delay factor $\alpha$ and filtration coefficient $\lambda$, (**b**) formation damage coefficient $\beta$, (**c**) initial attached concentration $\sigma_{aI}$, and (**d**) velocities $U_m$ and $U_i$.

Lastly, Figure 10d presents the sensitivity of the skin and breakthrough concentration with the maximum and critical velocities. Each velocity is scaled equally, effectively growing or shrinking the regions around the wellbore within which particle detachment occurs. When both velocities are high, both the region of total detachment (determined by $U_m$) and the region of any detachment (determined by $U_i$) are small, resulting in less overall detachment and lower skin/breakthrough concentration. On the other hand, when these velocities are small, detachment occurs in a larger region around the wellbore, and more fines are detached and undergo straining, resulting in more formation damage and more produced fines.

## 6. Treatment of Well Productivity Data

This section presents the history matching and forecast of skin growth for 10 production wells. The field data treatment and skin forecast are performed by tuning: (i) only the formation damage coefficient $\beta$ where linear skin-factor growth is observed (Cases 8–10) and (ii) the formation damage coefficient $\beta$, filtration coefficient $\lambda$, and drift-delay factor $\alpha$ where skin shows at least some tendency towards stabilisation (Cases 1–7). This deviation arises from the inability to uniquely identify more than one parameter from data which

falls on a straight line with fixed intercept. Thus, when the data shows no curvature, only one parameter is fit, and otherwise, all three can be identified.

Table 2 shows typical values for the assumed model parameters as well as the corresponding references. Table 3 shows the reference for each of the sets of well data, and Tables 4 and 5 show the fitting parameters and formation damage statistics, respectively. The fitting was done in Matlab using the solution presented in Appendix A. The integral terms were calculated numerically using the trapezoidal method. The fitting was done using the in-built linear least-squares solver utilizing a Levenberg-Marquadt algorithm. Figures 11–13 show plots of skin growth history matching and modelling for Cases 3, 7, and 9.

The parameters of the critical retention function are assumed in all cases because insufficient information is provided in the production data. For the wells studied in this work, only productivity index/skin data is given. The parameterisation of the critical retention function is possible when the time series of produced fines concentration is also available. An alternative approach is to combine coreflooding and field measurements, which were not available for the wells considered in this study.

**Table 2.** Typical values of fines-migration formation-damage coefficients.

| Parameters | Typical Values | Unit | References |
|---|---|---|---|
| $\sigma_{aI}$ | $5 \times 10^{-3}$ | - | Russell et al., 2017 [42] |
| $\lambda$ | $1 \times 10^{-2}$ | 1/m | Marquez et al., 2014 [43] |
| $U_m$ | $1 \times 10^{-4}$ | m/s | You et al., 2019 [24] |
| $\alpha$ | $1 \times 10^{-4}$–$1 \times 10^{-3}$ | - | Yang et al., 2016 [35] |

**Table 3.** The literature sources for production-well histories under fines migration.

| Case # | References |
|---|---|
| 1 | Marquez et al., 2014 [43] |
| 2 | Marquez et al., 2014 [43] |
| 3 | Kamps et al., 2010 [4] |
| 4 | Reinoso et al., 2016 [44] |
| 5 | Deskin et al., 1991 [45] |
| 6 | Ziauddin et al., 2002 [46] |
| 7 | Davidson et al., 1997 [47] |
| 8 | Marquez et al., 2014 [43] |
| 9 | Saldungaray et al., 2001 [5] |
| 10 | Marquez et al., 2014 [43] |

**Table 4.** Tuned and assumed values for fines-migration formation-damage model coefficients.

| Case # | $\sigma_{aI}$ | $U_m$ (m/s) | $U_i$ (m/s) | $\lambda$ (1/m) | $\alpha$ | $\beta$ | $R^2$ |
|---|---|---|---|---|---|---|---|
| 1 | $5 \times 10^{-3}$ | $1 \times 10^{-4}$ | $7 \times 10^{-5}$ | $2.3 \times 10^{-2}$ | $1.4 \times 10^{-3}$ | $7.56 \times 10^3$ | 0.912 |
| 2 | $5 \times 10^{-3}$ | $1 \times 10^{-4}$ | $7 \times 10^{-5}$ | $5 \times 10^{-2}$ | $1.3 \times 10^{-3}$ | $1.87 \times 10^3$ | 0.928 |
| 3 | $5 \times 10^{-3}$ | $1 \times 10^{-4}$ | $7 \times 10^{-5}$ | $2 \times 10^{-3}$ | $9.16 \times 10^{-5}$ | $2.80 \times 10^5$ | 0.966 |
| 4 | $5 \times 10^{-3}$ | $1 \times 10^{-4}$ | $7 \times 10^{-5}$ | $2.2 \times 10^{-2}$ | $7.6 \times 10^{-2}$ | $1.83 \times 10^3$ | 0.862 |
| 5 | $5 \times 10^{-3}$ | $1 \times 10^{-4}$ | $7 \times 10^{-5}$ | $1.2 \times 10^{-2}$ | $1.7 \times 10^{-3}$ | $3.3 \times 10^3$ | 0.963 |
| 6 | $5 \times 10^{-3}$ | $1 \times 10^{-4}$ | $7 \times 10^{-5}$ | $1 \times 10^{-2}$ | $7 \times 10^{-5}$ | $7.57 \times 10^4$ | 0.864 |
| 7 | $5 \times 10^{-3}$ | $1 \times 10^{-4}$ | $7 \times 10^{-5}$ | $1 \times 10^{-2}$ | $7 \times 10^{-5}$ | $8.78 \times 10^4$ | 0.984 |
| 8 | $5 \times 10^{-3}$ | $1 \times 10^{-4}$ | $7 \times 10^{-5}$ | $1 \times 10^{-2}$ | $1 \times 10^{-3}$ | $7.83 \times 10^3$ | 0.949 |
| 9 | $5 \times 10^{-3}$ | $1 \times 10^{-4}$ | $7 \times 10^{-5}$ | $1 \times 10^{-2}$ | $1 \times 10^{-4}$ | $1.48 \times 10^4$ | 0.967 |
| 10 | $5 \times 10^{-3}$ | $1 \times 10^{-4}$ | $7 \times 10^{-5}$ | $1.1 \times 10^{-2}$ | $3 \times 10^{-4}$ | $3.55 \times 10^4$ | 0.917 |

**Table 5.** Prediction of damaged-well exploitation parameters based on the well production history.

| Case # | $r_e$ (m) | $r_w$ (m) | $r_m$ (m) | $r_i$ (m) | $r_d$ (m) | $J_{stab}$ | $S_{stab}$ | $T_{stab}$ (PV) |
|--------|-----------|-----------|-----------|-----------|-----------|------------|------------|-----------------|
| 1 | 1000 | 0.1 | 5.39 | 7.71 | 0.80 | 6.57 | 50.62 | 0.040 |
| 2 | 1000 | 0.1 | 5.39 | 7.71 | 0.82 | 3.72 | 25.17 | 0.043 |
| 3 | 1000 | 0.1 | 1.84 | 2.63 | 0.58 | 2.69 | 15.6 | 0.072 |
| 4 | 1000 | 0.1 | 4.48 | 6.40 | 0.76 | 1.41 | 3.74 | 0.041 |
| 5 | 1000 | 0.1 | 8.89 | 12.71 | 0.86 | 4.29 | 30.37 | 0.09 |
| 6 | 1000 | 0.1 | 1.84 | 2.63 | 0.58 | 3.61 | 24.09 | 0.094 |
| 7 | 1000 | 0.1 | 1.84 | 2.63 | 0.58 | 4.01 | 27.81 | 0.33 |
| 8 | 1000 | 0.1 | 34.69 | 49.56 | 0.97 | 90.26 | 822.1 | 2.26 |
| 9 | 1000 | 0.1 | 3.91 | 5.59 | 0.74 | 3.38 | 22 | 0.31 |
| 10 | 1000 | 0.1 | 1.84 | 2.63 | 0.58 | 2.34 | 12.38 | 0.023 |

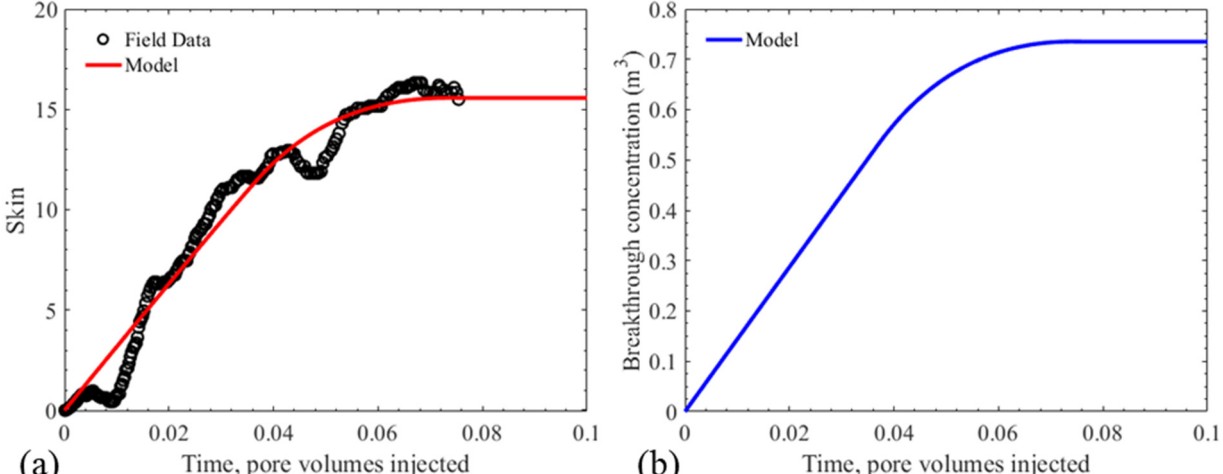

**Figure 11.** Results of tuning three model coefficients (filtration and formation damage coefficients, and drift delay factor) for a well during the later stages of skin growth (Case 3 from Kamps, Chando and Ellis [4]), (**a**) skin vs. PVI, (**b**) the cumulative fines production at the wellbore vs. PVI.

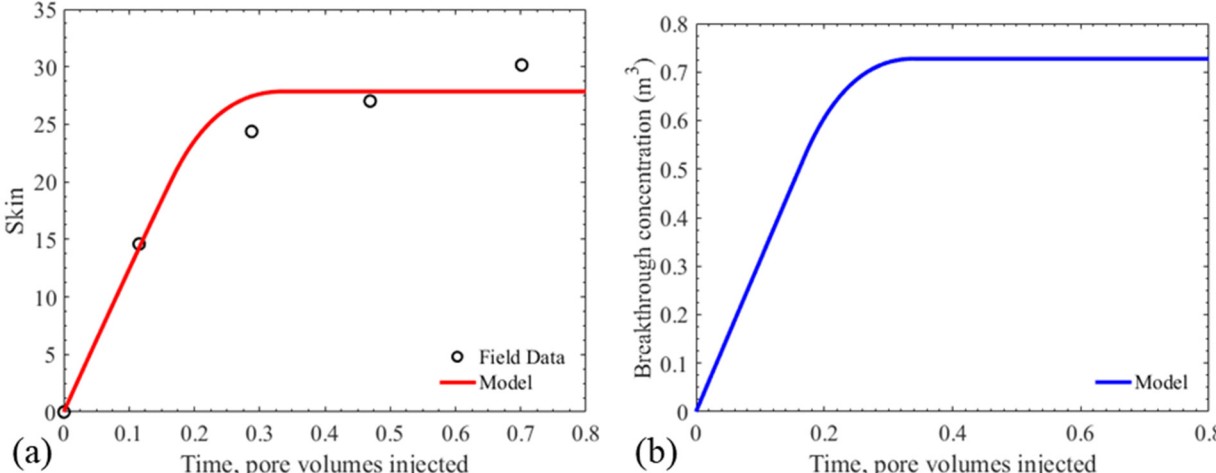

**Figure 12.** Results of tuning three model coefficients (filtration and formation damage coefficients, and drift delay factor) for a well during the later stages of skin growth (Case 7 from Davidson, Franco, Gonzalez and Robinson [47]), (**a**) skin vs. PVI, (**b**) the cumulative fines production at the wellbore vs. PVI.

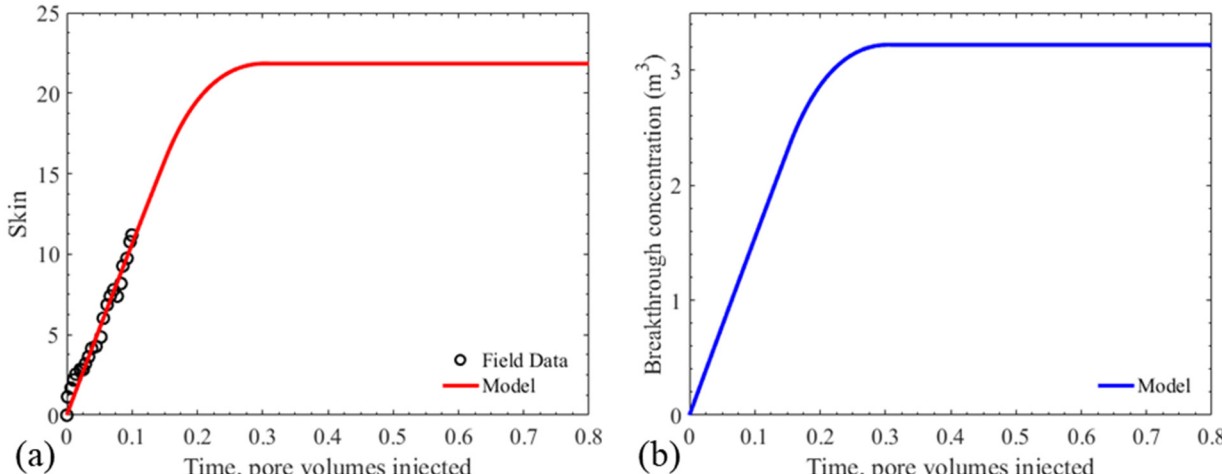

**Figure 13.** Results of tuning one model coefficient (formation damage coefficient) for a well during the early stages of skin growth (Case 9 from Saldungaray, Caretta, and Sofyan [5]), (**a**) skin vs. PVI, (**b**) the cumulative fines production at the wellbore vs. PVI.

Figure 11 shows the growth of the skin factor against the total produced oil, presented as pore volumes (PV). For this data, oil is produced from six Upper Cretaceous fields in the Doba Basin in Chad, Central Africa [4]. The rock is an extremely unconsolidated sandstone consisting of 4% fine content, of which more than 90% have been identified as kaolinite. During a production period of 15 months, the well productivity index decreased 2.8 times, and the authors of this study identified fines migration as the primary cause of the formation damage. Given that the trend in the data is non-linear, all three model parameters ($\beta$, $\lambda$, $\alpha$) are tuned. Other parameters present in the model ($\sigma_{aI}$, $U_m$, and $U_i$) are chosen from commonly reported values in other studies, as presented in Table 2. The final values for all parameters are presented in Table 4 (Case 3), and a summary of the formation damage statistics are presented in Table 5. Within a 1.84 m neighbourhood of the well ($r_m$), all fines are mobilised at the beginning of production. Beyond 2.63 m from the wellbore, no fines detachment occurs ($r_i$), and between these two values, partial detachment is predicted. The formation damage radius, $r_d$, is defined as the radius within which almost all meaningful formation damage occurs. Thus, if all strained fines within this region were removed, the initial productivity would be approximately restored. For this well, the formation damage radius is 0.58 m. The smaller value of the damage radius compared to $r_i$ and $r_m$ reflects the distance travelled by particles towards the wellbore between detachment and straining, the higher intensity of straining near the wellbore due to high velocities, and the higher relative importance of rock permeabilities closer to the wellbore for the overall well productivity.

The tuned model exhibits a decrease in the well index of 2.69 times during 0.072 PV, corresponding to a skin factor of 15.6. The coincidence between the well index decrease from the model and data is consistent with the overall high quality of fit, as seen in Figure 11. The model accurately captures the trend of the data, although the smooth model cannot capture the fluctuations away from the trend.

The next case presented is Case 7, shown in Figure 12. This data is from a well in the Toroyaco field located in the Putumayo Basin in Southern Colombia. The main producing intervals are the Villeta U and T Sandsas reported by Davidson, Franco, Gonzalez and Robinson [47]. During the 6 years of recorded production, the well productivity index decreased 4.27 times. Similar to the previous case, the formation damage coefficient, filtration coefficient, and drift delay factor all fit given the non-linearity of the data. The other three parameters, the initial concentration of movable fines $\sigma_{aI}$, critical velocity $U_i$, and maximum velocity $U$, are assumed (see Table 4). The model exhibits stabilisation of the well productivity after 0.33 PV of produced oil, at which point the production rate has decreased by a factor of 4.01, corresponding to a skin value of 27.81. These numbers are

consistent with the production data, and the model agreement with the data is overall very good. For this well, total detachment occurs within a radius of 1.84 m around the well, and partial detachment continues until 2.63 m. The 4-fold decrease in productivity is a result of strained particles within a radius of 0.58 m around the well.

For Case 9, the data is taken for a well located in the South East Sumatra field offshore Indonesia [5]. This field is located in the Java Sea. The reservoir rocks are sandstones, with clay content varying between 1–7%. During the 17 months of production, the well productivity index decreased 2.2 times. As shown in Figure 13, the skin growth for this well is approximately linear, with no significant tendency towards stabilisation, indicating that the well is in the early stages of damage due to fines migration. As such, tuning is performed only for the formation damage coefficient, $\beta$. As will be discussed later, without a viable means to treat all three parameters ($\beta$, $\lambda$, $\alpha$), predicting the stabilisation time is not possible. Given the assumed values of $\lambda$ and $\alpha$, a numerical prediction is made in Figure 13; however, this prediction is subject to the uncertainties associated with using average values from the literature. Nonetheless, the curve can provide at least order of magnitude estimates that can be useful for field operations. For this data, the model predicts the stabilisation of skin growth after the production of 0.31 PV of oil. The final value of the skin is 22, which corresponds to a 3.38-fold decrease in the well index. The size of the damaged zone is 0.74 m.

## 7. Discussion

### 7.1. Unique Determination of the Six Model Coefficients

The analytical model (A1 to A17) allows for the calculation of pressure drawdown and well index through explicit calculations of suspended and strained particle concentrations during axi-symmetric flow towards a production well. It can be readily implemented in Excel or Matlab, which makes it a practical tool for reservoir and production engineers.

The model consists of two main components: the description of particle detachment and the modelling of particle transport and capture. The model for particle detachment involves three parameters: the initial concentration of attached fines, $\sigma_{aI}$, the critical velocity, $U_i$, and the maximum velocity, $U_m$. These parameters are required to define the critical retention function, $\sigma_{cr}(U)$ (Equation (37)). The model for suspension-colloidal transport also contains three parameters: the formation damage coefficient, $\beta$, the filtration coefficient, $\lambda$, and the drift delay factor, $\alpha$. Thus, when modelling fines migration in porous media using the developed analytical model (A1 to A17), we require the determination of six parameters. Unique determination of this many parameters may require several sets of data, such as pressure drops, concentrations, or deposit profiles. Given the difficulty of obtaining this data in field scenarios, this analysis is mostly restricted to laboratory corefloods. Therefore, the complete characterisation of the fines migration system during well production based on the production history can only be performed in combination with laboratory coreflood data.

In the absence of laboratory coreflood tests, well production data can be used to match "skin-factor vs. time" using the analytical model (A1 to A17) for the following two scenarios:

I.    At the beginning of production and the initial stage of well exploitation, skin $S(T)$ grows linearly with time from its initial value of zero. Therefore, only one parameter can be determined from the skin curve $S(T)$. The other five parameters must be chosen from values commonly reported in the literature;

II.    When the skin curve shows a tendency towards stabilisation, three parameters can be determined from the skin vs. time, and the other three parameters must be chosen from values commonly reported in the literature.

Following the tuning procedures outlined above, the model parameters $\lambda$, $\sigma_{aI}$, $U_i$, and $U_m$, allow for the calculation of the radius of the damaged zone, $r_d$. Determination of the size of the damaged zone $r_d$ allows planning of well stimulation, such as selecting the necessary perforation length or the required volume of acid. Since the remediation of the formation damage corresponds to the removal of strained particles in the damaged zone,

the minimum volume of acid required to restore well productivity is $\pi(r_d{}^2 - r_w{}^2)\phi$ (per unit formation height). In the case of well perforation and fracturing, in order to bypass the damaged rock, the length of perforation must exceed $r_d$. The values of the damaged zone calculated in this study do not exceed 1 m (see Table 5), indicating that well remediation is reasonable for the wells studied in this work.

For chemical treatments designed to prevent fines migration, the design should be based on the radius, $r_i$, outside of which there is no detachment. Injection of treating fluids such as polymers [5] or oil [2] should aim to inject a sufficient volume of treating fluid to treat attached fines out to this critical radius, as fines beyond this radius are not expected to detach due to low fluid velocities.

*7.2. Incorporating More Complex Reservoir Physics*

In the model in this paper, we have assumed that particle detachment only occurs due to the drag force exerted by the oleic phase on the particles, thus neglecting any detachment of particles in pores filled with connate water. It has been reported elsewhere that these particles are not subject to detachment during oil flow [48,49]. The model presented here quantifies the fraction of internal pore surface covered by the connate water, although more complex investigations have been done, showing the importance of rock wettability and surface roughness [50,51]. In this context, the simplified physical picture used in this model of irreducible water completely filling some subset of pores is a disadvantage, as it neglects more complex configurations such as water films [52].

Another disadvantage of this work is that, unlike in many oil production scenarios, only one phase is mobile. When both phases are mobile, each phase could detach particles. These conditions highlight the importance of the oil-water meniscus, which could play a vital role both in particle detachment and transport. The capillary force exerted by the meniscus on attached particles has been shown to greatly exceed both the electrostatic and drag forces [26]. In this case, modelling formation damage due to fines migration would require explicit modelling of the oil-water interface, as determined by the equations of transport of the separating surface during two-phase flow [12,13]. In addition to the effect of the meniscus, in the presence of gradients in ionic composition, cation exchange on the rock and particle surfaces can induce fines mobilisation [53]. These phenomena influencing the detachment of fines are important considerations during oil and gas production [54,55] and low-salinity or smart water injection [30,51,56–59].

An advantage of the model in this work is that it is sufficiently simple to allow an analytical solution. Additional complexity can be incorporated without sacrificing this by performing upscaling. For example, a large variety of particle and pore sizes can be incorporated into stochastic models for suspension-colloidal transport in porous media [29,60,61]. Another example is extensions to fines capture by noting the similarities between the kinetics of fines straining and chemical reactions [62,63]. These systems allow for more physically accurate models but allow for exact upscaling and can thus extend the analytical model presented in this work. While we have used a constant for the surface coverage parameter $\gamma$ in Equation (2), another advantage of the model is the easy application of a more complex function. It follows from intuitive arguments that particles that appear due to the chemical transformation of existing material in the porous media (authigenic particles) might be evenly distributed across the pore space or more concentrated in smaller, clay-bearing pores. On the other hand, particles that previously detached and have been deposited within the pore space during deposition or diagenesis (allogenic/detrital particles) may be more concentrated in larger pores where the flux is greater. These considerations can lead to a wide variety of functions $\gamma(r_p)$ that can be easily implemented into the model.

This work focuses only on formation damage due to fines migration. In some wells, there are multiple sources of formation damage. For these cases, the skin factor for each mechanism can be added, resulting in a total skin. In this way, the Equations used in this work can be used to account for the influence of fines migration.

The Equations can be adopted to model the case of mobile water and immobile oil. This case is less common but can be encountered during tertiary waterflooding in oil reservoirs. In this situation, rather than neglecting particles in the small, water-filled pores ($r < r_c$), we would neglect the largest pores ($r > r_c$) in Equation (3) which contain the immobile oil.

### 8. Summary and Conclusions

The analytical study of fines production during oil production in reservoirs with connate water allows drawing the following conclusions:

- A new form of the critical retention function is derived based on a pore space comprised of a size-distributed bundle of capillaries. The new formulation allows for including the effects of connate water on fines detachment during oil production;
- Connate water saturation can significantly decrease maximum retention function by preventing fines detachment from the pores filled by the immobile water;
- The new equations show that skin growth is more severe in reservoirs with low connate water saturation, where more particles can be detached by the mobile oil phase;
- The axi-symmetric flow of oil towards a well with the mobilisation, transport, and straining of fine particles under the presence of connate water allows for an analytical solution. In the case of a large concentration of suspended particles, where the retention rate is proportional to the suspension function, $F(C)$, the expressions for the suspended and strained particle concentrations are implicit. For the two cases where the function $F(C)$ is quadratic or for the case where the suspended concentration is sufficiently small to assume $F(C) = C$, the expressions are explicit;
- The analytical model allows the quantification of the growth of well impedance and skin during fines mobilization;
- Analysis of 10 field production wells shows good agreement between the data and the analytical model. The formation damage parameters obtained from tuning are within commonly reported intervals;
- The final model contains six parameters which describe the extent of fines migration. Depending on whether the skin history curve covers the initial productivity decline or if it contains the progression towards productivity stabilisation, the curve can be used to determine 1–3 parameters. Thus the full determination of the system requires laboratory coreflooding. An alternative approach is to assume typical values of a subset of the parameters from published research in similar rocks;
- Following tuning, the analytical model provides accurate estimates of the well skin growth until stabilization, including the final skin value, as well as the size of the formation damage zone around the well. This information allows field operators to make informed decisions on well design and well-stimulation procedures.

**Author Contributions:** Conceptualization, L.C., T.R., A.Z. and P.B.; Formal analysis, G.L., C.N., T.R. and P.B.; Investigation, G.L. and L.C.; Methodology, G.L., C.N. and T.R.; Supervision, L.C., A.Z. and P.B.; Validation, C.N.; Writing—original draft, P.B.; Writing—review and editing, T.R. and A.Z. All authors have read and agreed to the published version of the manuscript.

**Funding:** This research received no external funding.

**Data Availability Statement:** No new data were created or analyzed in this study. Data sharing is not applicable to this article.

**Conflicts of Interest:** The authors declare no conflict of interest.

### Nomenclature

Latin letters
$c$      Suspended particles concentration
$C$      Dimensionless suspended particles concentration
$C_v$      Coefficient of variation of the pore size distribution
$J$      Impedance

| | |
|---|---|
| $k$ | Permeability, $[L]^2$, $m^2$ |
| $M_{rp}$ | Mean pore size, $[L]$, m |
| $p$ | Pressure, $[M][T]^{-2}[L]^{-1}$, Pa |
| $P$ | Dimensionless pressure |
| $q$ | Flow rate per unit of the reservoir thickness, $[L]^2[T]^{-1}$, $m^2s^{-1}$ |
| $r$ | Radial coordinate, $[L]$, m |
| $r_e$ | Drainage radius, $[L]$, m |
| $r_i$ | Radius of the zone where particles are detached, $[L]$, m |
| $rm$ | Radius of the zone where all attached particles are detached, $[L]$, m |
| $s$ | Saturation |
| $S$ | Skin factor |
| $S_a$ | Dimensionless concentration of attached particles |
| $S_s$ | Dimensionless concentration of strained particles |
| $t$ | Time, $[T]$, s |
| $T$ | Dimensionless time, PVI |
| $U$ | Darcy's velocity, $[L][T]^{-1}$, $m.s^{-1}$ |
| $U_i$ | Darcy's velocity corresponding to $r = r_i$, $[L][T]^{-1}$, $m.s^{-1}$ |
| $U_m$ | Darcy's velocity corresponding to $r = r_m$, $[L][T]^{-1}$, $m.s^{-1}$ |
| $X$ | Dimensionless radial coordinate |
| $X_i$ | Dimensionless radius of the zone where particles are detached |
| $X_m$ | Dimensionless radius of the zone where all attached particles are detached |
| $X_0$ | Intersection of characteristic line with $x$ axis |
| Greek letters | |
| $\alpha$ | Drift delay factor |
| $\beta$ | Formation damage coefficient |
| $\varepsilon$ | Accuracy |
| $\gamma$ | Salinity |
| $\lambda$ | Filtration coefficient, $[L]^{-1}$, $m^{-1}$ |
| $\Lambda$ | Dimensionless filtration coefficient |
| $\mu$ | Dynamic viscosity, $[M][L]^{-1}[T]^{-1}$, $kg.m^{-1}s^{-1}$ |
| $\sigma_a$ | Concentration of attached particles |
| $\sigma_{aI}$ | Initial attached particles concentration |
| $\sigma_{a0}$ | Concentration of attached particles for $U = 0$ m/s |
| $\sigma_s$ | Concentration of strained particles |
| $\phi$ | Porosity |
| $\omega$ | Drag coefficient |
| tSuper/subscripts | |
| $cr$ | Critical, retention concentration |
| $d$ | Damage, for radius |
| $w$ | *Well, for pressure and radius* |
| $wi$ | Water initial (for end point) |

## Appendix A. Exact Solution for Linear Suspension Function $F(C) = C$

In the case of low suspension concentrations, for a simple expression (9) for the suspension function, the solution (28) allows for explicit expressions. Below we obtain those expressions $C(X,T)$ from Equation (28) and $F(C) = C$ in zones 1, 2–3, and 4–6. Following the integration-in-$T$ procedure presented in the previous session, below, we calculate $S_s(X,T)$ in six zones.

$$\text{Zone 1 } \left( X_w < X + \alpha(1 - s_{wi})^{-1}T < X_m \right) \tag{A1}$$

$$C(X, T) = e^{\Lambda\left( \sqrt{X} - \sqrt{X + \alpha(1 - s_{wi})^{-1}T} \right)} \tag{A2}$$

$$S_s(X, T) = \frac{(1 - s_{wi})}{\Lambda\sqrt{X}} \left[ \Lambda\sqrt{X} + 1 - \left( \Lambda\sqrt{X + \alpha(1 - s_{wi})^{-1}T} + 1 \right) e^{\Lambda\left( \sqrt{X} - \sqrt{X + \alpha(1 - s_{wi})^{-1}T} \right)} \right] \tag{A3}$$

$$\text{Zone 2 } \left( X_w < X < X_m, X_m < X + \alpha(1-s_{wi})^{-1}T < X_i \right) \tag{A4}$$

$$C(X,T) = \left( 1 - S_{cr}\left( \frac{q}{2\pi r_e \sqrt{X+\alpha T}} \right) \right) e^{\Lambda\left( \sqrt{X} - \sqrt{X+\alpha(1-s_{wi})^{-1}T} \right)} \tag{A5}$$

$$S_s(X,T) = \frac{(1-s_{wi})}{\Lambda\sqrt{X}}\left[ \Lambda\sqrt{X} + 1 - \left( \Lambda\sqrt{X_m} + 1 \right)e^{\Lambda(\sqrt{X}-\sqrt{X_m})} \right]$$

$$+ \frac{(\sigma_{aI} - \sigma_{a0})(1-s_{wi})}{\sigma_{aI}\Lambda\sqrt{X}}\left[ \left( \Lambda\sqrt{X_m}+1 \right)e^{\Lambda(\sqrt{X}-\sqrt{X_m})} - \left( \Lambda\sqrt{X+\alpha(1-s_{wi})^{-1}T} + 1 \right)e^{\Lambda\left( \sqrt{X} - \sqrt{X+\alpha(1-s_{wi})^{-1}T} \right)} \right] \tag{A6}$$

$$+ \frac{\alpha\Lambda\sigma_{a0}q^2}{8\pi^2 r_e^2 U_m^2 \sigma_{aI}\sqrt{X}} e^{\Lambda\sqrt{X}} \int_{\frac{X_m-X}{\alpha}}^{T} \frac{1}{X+\alpha(1-s_{wi})^{-1}T} e^{-\Lambda\sqrt{X+\alpha(1-s_{wi})^{-1}T}} dT$$

$$\text{Zone 3 } \left( X_m < X < X_i, X + \alpha(1-s_{wi})^{-1}T < X_i \right) \tag{A7}$$

$$C(X,T) = \left( 1 - S_{cr}\left( \frac{1}{2\pi r_e \sqrt{X+\alpha T}} \right) \right) e^{\Lambda\left( \sqrt{X} - \sqrt{X+\alpha(1-s_{wi})^{-1}T} \right)} \tag{A8}$$

$$S_s(X,T) = \frac{(\sigma_{aI} - \sigma_{a0})(1-s_{wi})}{\sigma_{aI}\Lambda\sqrt{X}}\left[ \Lambda\sqrt{X} + 1 - \left( \Lambda\sqrt{X+\alpha(1-s_{wi})^{-1}T} + 1 \right)e^{\Lambda\left( \sqrt{X} - \sqrt{X+\alpha(1-s_{wi})^{-1}T} \right)} \right]$$

$$+ \frac{\alpha\Lambda\sigma_{a0}q^2}{8\pi^2 r_e^2 U_m^2 \sigma_{aI}\sqrt{X}} e^{\Lambda\sqrt{X}} \int_{0}^{T} \frac{1}{X+\alpha(1-s_{wi})^{-1}T} e^{-\Lambda\sqrt{X+\alpha(1-s_{wi})^{-1}T}} dT \tag{A9}$$

$$\text{Zone 4 } \left( X_w < X < X_m, X + \alpha(1-s_{wi})^{-1}T > X_i \right) \tag{A10}$$

$$C(X,T) = 0 \tag{A11}$$

$$S_s(X,T) = \frac{(1-s_{wi})}{\Lambda\sqrt{X}}\left[ \Lambda\sqrt{X} + 1 - \left( \Lambda\sqrt{X_m}+1 \right)e^{\Lambda(\sqrt{X}-\sqrt{X_m})} \right] + \frac{(\sigma_{aI}-\sigma_{a0})(1-s_{wi})}{\sigma_{aI}\Lambda\sqrt{X}}\left[ \begin{array}{c} \left(\Lambda\sqrt{X_m}+1\right)e^{\Lambda(\sqrt{X}-\sqrt{X_m})} \\ -\left(\Lambda\sqrt{X_i}+1\right)e^{\Lambda(\sqrt{X}-\sqrt{X_i})} \end{array} \right]$$

$$+ \frac{\alpha\Lambda\sigma_{a0}q^2}{8\pi^2 r_e^2 U_m^2 \sigma_{aI}\sqrt{X}} e^{\Lambda\sqrt{X}} \int_{\frac{X_m-X}{\alpha}}^{\frac{X_i-X}{\alpha}} \frac{1}{X+\alpha(1-s_{wi})^{-1}T} e^{-\Lambda\sqrt{X+\alpha(1-s_{wi})^{-1}T}} dT \tag{A12}$$

$$\text{Zone 5 } \left( X_m < X < X_i, X + \alpha(1-s_{wi})^{-1}T > X_i \right) \tag{A13}$$

$$C(X,T) = 0 \tag{A14}$$

$$S_s(X,T) = \frac{(\sigma_{aI}-\sigma_{a0})(1-s_{wi})}{\sigma_{aI}\Lambda\sqrt{X}} e^{\Lambda\sqrt{X}}\left[ \Lambda\sqrt{X} + 1 - \left( \Lambda\sqrt{X_i}+1 \right)e^{\Lambda(\sqrt{X}-\sqrt{X_i})} \right]$$

$$+ \frac{\alpha\Lambda\sigma_{a0}q^2}{8\pi^2 r_e^2 U_m^2 \sigma_{aI}\sqrt{X}} e^{\Lambda\sqrt{X}} \int_{0}^{\frac{X_i-X}{\alpha}} \frac{1}{X+\alpha(1-s_{wi})^{-1}T} e^{-\Lambda\sqrt{X+\alpha(1-s_{wi})^{-1}T}} dT \tag{A15}$$

$$\text{Zone 6 } (X > X_i) \tag{A16}$$

$$C(X,T) = S_s(X,T) = 0 \tag{A17}$$

## Appendix B. Derivation of Critical Retention Function with Connate Water

Suppose the porous media consists of a bundle of parallel cylindrical capillaries of equal length, $l$. The capillaries are distributed by their radius according to some distribution function, $f(r_p)$. On the internal surface of each capillary is an evenly distributed monolayer of monosized attached particles, which covers a $\gamma(r_p)$ fraction of the surface area. Thus the total internal pore area covered by particles in a single pore is:

$$2\pi r_p l \gamma(r_p) \tag{A18}$$

The number of particles is found by dividing by the projected area of a particle:

$$\frac{2r_p l \gamma(r_p)}{r_s^2} \tag{A19}$$

Multiplying by the particle volume yields the volume of the particles

$$\frac{8}{3}\pi r_p r_s l \gamma(r_p) \tag{A20}$$

The volume of the pore is

$$\pi l r_p^2 \tag{A21}$$

Thus integrating the particle volume and pore volume separately over the pore size distribution and dividing results in the critical retention function:

$$\sigma_{cr} = \frac{V_{particles}}{V_b} = \frac{8}{3}\phi r_s \frac{\int_0^\infty \gamma(r_p) r_p f(r_p) dr_p}{\int_0^\infty r_p^2 f(r_p) dr_p} \tag{A22}$$

To account for the velocity and connate water saturation dependencies, we introduce two critical pore radii, $r_c(U)$ and $r_c(S_{wi})$, defined in Equations (3) and (4).

Incorporating these two effects involved removing all particles from pores with radii larger than $r_c(U)$ and preventing the detachment of particles in pores with radii less than $r_c(S_{wi})$.

**Appendix C. Evaluation of the Torque Balance with Distributed Pore Radii**

For the assembly of parallel cylindrical pores, we can compute the flux through each pore using Stokes' law:

$$q = -\frac{\Delta p \pi r_p^4}{8\mu L} \tag{A23}$$

The total flux through all pores is given by

$$q_t = \frac{\Delta p \pi}{8\mu L} \int_0^\infty r_p^4 f(r_p) dr_p \tag{A24}$$

Dividing the two equations and rearranging results in

$$q = \frac{r_p^4 q_t}{\int_0^\infty r_p^4 f(r_p) dr_p} \tag{A25}$$

The total flux can be replaced by the average pore velocity, $U_p$, by dividing by the pore cross-sectional area

$$U_p = \frac{r_p^2 q_t}{\pi \int_0^\infty r_p^4 f(r_p) dr_p} \tag{A26}$$

Similarly, for the total flux

$$q_t = U \int_0^\infty \pi r_p^2 f(r_p) dr_p \tag{A27}$$

Thus

$$U_p = \frac{r_p^2 U \int_0^\infty r_p^2 f(r_p) dr_p}{\int_0^\infty r_p^4 f(r_p) dr_p} \tag{A28}$$

Within each pore, the velocity varies along the distance, $r$, from the central axis as per

$$U(r) = \frac{\Delta p}{4\mu L}\left(r^2 - r_p^2\right) \tag{A29}$$

Using Stokes' law and replacing the flux with the average pore velocity:

$$U(r) = 2U_p\left(1 - \frac{r^2}{r_p^2}\right) \tag{A30}$$

If a particle is attached to the internal pore surface, then the velocity acting at its centre is as follows

$$U(r_p - r_s) = 2U_p\left(1 - \frac{(r_p - r_s)^2}{r_p^2}\right) \tag{A31}$$

The drag force acting on the particle is [64]:

$$F_d = 6\pi\mu r_s U(r_p - r_s)\omega \tag{A32}$$

where $\mu$ is the fluid viscosity, and $\omega$ is the drag coefficient.

Substituting Equation (A31) for the velocity acting on the particle:

$$F_d = 12\pi\mu r_s U_p\left(1 - \frac{(r_p - r_s)^2}{r_p^2}\right)\omega \tag{A33}$$

Substituting the average pore velocity as given by Equation (A28) results in:

$$F_d(U_p, r_s) = 12\pi\mu r_s \omega r_p^2 U\left(1 - \frac{(r_p - r_s)^2}{r_p^2}\right)\frac{\int_0^\infty r_p^2 f(r_p) f r_p}{\int_0^\infty r_p^4 f(r_p) dr_p} \tag{A34}$$

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
