# Peer review of "Treatment of Oil Production Data under Fines Migration and Productivity Decline"

_energies, doi:10.3390/en16083523_

Round 1

Reviewer 1 Report

The presented manuscript deals with a derivation of a new model for fine particle detachment for calculations of skin growth and productivity decline in subterranean reservoirs.

The authors well introduced the research topic and the current limitations of productivity decline modeling. The introduced model is well described including its limitations and advantages. 

I'm in favor of publishing the present manuscript.

Several typo errors were found. (They might be associated with a PDF reader)

Line 355-356: Wrong symbols for attached and strained concentrations a porosity.

Line 465-468, 487, 494: use of subscripts for Xw, Xi, Xm.

Line 600-616: use of subscripts. 

Line 677-678: Wrong symbol for the formation damage coefficient.

Line 765: Wrong symbol.

Line 818: Wrong symbol for the critical retention function.

Line 845: Wrong symbol.

Author Response

Author Response to Reviewer 1

The presented manuscript deals with a derivation of a new model for fine particle detachment for calculations of skin growth and productivity decline in subterranean reservoirs.

The authors well introduced the research topic and the current limitations of productivity decline modeling. The introduced model is well described including its limitations and advantages. 

I'm in favor of publishing the present manuscript.

Several typo errors were found. (They might be associated with a PDF reader)

Yes, we noticed that the conversion to the journal format induced several formatting errors. These have been fixed throughout the text.

Line 355-356: Wrong symbols for attached and strained concentrations a porosity.

Fixed.

Line 465-468, 487, 494: use of subscripts for Xw, Xi, Xm.

Fixed.

Line 600-616: use of subscripts.

Fixed.

Line 677-678: Wrong symbol for the formation damage coefficient.

Fixed.

Line 765: Wrong symbol.

Fixed.

Line 818: Wrong symbol for the critical retention function.

Fixed.

Line 845: Wrong symbol.

Fixed.

Reviewer 2 Report

In this paper, authors have built a model to predict skin grown due to fine migration. I think authors have done a solid work and it can be considered for publication in Energies. Some questions and suggestions are listed as follows,

1. For the effect of pore size distribution, the normal distribution is used. Did authors try other distribution, the pore size distribution may be very complex.

2. The model is built based on the irreducible water phase. Can it be used in situation when water is mobile. Oil and water flow simultaneous may be more common situation.

3. It is good to see the model is verified by successful matching of 10 field cases. However, fines migration may be only one mechanism of skin growth. What do you think the applicability of this model for skin growth with complex mechanism?

Author Response

Author response to Reviewer 2

In this paper, authors have built a model to predict skin grown due to fine migration. I think authors have done a solid work and it can be considered for publication in Energies. Some questions and suggestions are listed as follows,

  1. For the effect of pore size distribution, the normal distribution is used. Did authors try other distribution, the pore size distribution may be very complex.

We agree that pore size distribution can be very complex. Our example calculation only includes a normal distribution but the derivation can be applied for any probability distribution. We have added some comments to highlight this, see page 9, lines 313-315.

  1. The model is built based on the irreducible water phase. Can it be used in situation when water is mobile. Oil and water flow simultaneous may be more common situation.

The presented model only applies when a single phase is mobile. The work is further developments of other single-phase works that don’t account for the influence of a second immobile phase. We agree that two mobile phases is common, and it will introduce more complex physics. We discuss these phenomena in page 25, lines 874-885.

  1. It is good to see the model is verified by successful matching of 10 field cases. However, fines migration may be only one mechanism of skin growth. What do you think the applicability of this model for skin growth with complex mechanism?

Agree. When multiple formation damage mechanisms are present we can add the skin induced by each mechanism, including the one developed in this work for fines migration. We have added some comments on this topic in the text, see page 26, lines 902-905.

Reviewer 3 Report

With the title “Treatment of oil production data under fines migration and productivity decline”, the paper appears suitable for publication in energies journal.

The authors address a practical problem faced by the oil and gas industry, the fine migration as the causes of the decline in oil production and well damage. The authors have investigated the particles migration and propose a novel mathematical model that accounts for immobile fines trapped within irreducible water saturation. The work is worthy for publication. Comments and suggestions can be found in the attached report.

Author Response

Author response to Reviewer 3

With the title “Treatment of oil production data under fines migration and productivity decline”, the paper appears suitable for publication in energies journal.

The authors address a practical problem faced by the oil and gas industry, the fine migration as the causes of the decline in oil production and well damage. The authors have investigated the particles migration and propose a novel mathematical model that accounts for immobile fines trapped within irreducible water saturation. The work is worthy for publication.

Comments and suggestions to the authors can be found in the attached are provided below:

  1. On the novelty of this work, the current model for particle detachment account for the immobile connate water. an consideration can be made on the immobile oil phase. can it affect the particle detachment?

Yes, the model could be modified to model the situation of mobile water with immobile oil. In this case, the part of the pore size distribution ‘neglected’ for the critical retention function should be the right tail (r>rc) as opposed to the left tail (r<rc) as we do in Eq. (3). We have included some discussions on this topic in the discussions, see page 26, lines 906-909.

  1. Any recommendation can be drawn to expand your work for when both phases, oil and water are mobile? If so, please place it into your paper.

Yes, this case is interesting, but introduces new physical effects. We discuss this in the paper in page 25, lines 874-885.

  1. Near the injection wells, the particle detachment may be high. how does this model cope with this?

The model accounts for higher detachment near the wellbore with a velocity dependent critical retention function, see Eq. (2) or Eq. (37) for a simple form, as well as the relationship between velocity and distance from the wellbore, Eq. (6). These two effects indeed result in more detachment, which we see in the initial suspended concentration in Eq. (11).

  1. As mentioned in the report, water salinity may contribute for particle detachment, what is the limitation of your study? can this model handle all types of water salinity?

Agree, salinity is an important factor in determining the extent of fines migration. In our model, the water is immobile, so a constant salinity is likely. The difference in fines migration between a reservoir with high salinity connate water vs low salinity will be captured by the critical retention function. This will affect the electrostatic force in Eq. (5), changing rc(U), leading to salinity dependent detachment.

-L30-31: insert required number of keywords

Done.

-L152-153: misspelling “assumptes”

Fixed.

-L152-153: misspelling “pre”

Fixed, we meant to say “pore”.

-L266 and 278: please insert a full stop mark

We do not use full stop at the end of figure caption in line with this journals formatting

L284-284: please review syntax

Agree, the syntax was unclear, it has been fixed.

L354-356: please check your symbols used in these lines to be corrected

Some formatting issues occurred during conversion by the journal to their format. They have been fixed throughout the text.

L460 and 583: review symbols representation, e.g. X0

See above comment.

L438: please review the terminology used in your equations T may have another meaning than t in the equation

In this work, t is the time, and T is the dimensionless time, defined in Eq. (15).

L822-823: please review sentence syntax

Done.

L845: Is the minimum volume of acid formula correct?

The formatting error seems to have removed the porosity factor. This calculation is only an estimation and more rigorous analysis is required to properly design an acidizing job.

L850-851: in this line, is critical radii denoted by ri or rc?

The reference to ri is correct, but the wording is confusing, it has been changed.

L1024: references appear to not follow the journal guide. reference 1, there is a mistype

Fixed.

Reviewer 4 Report

Some figures need attention in terms of quality and proportionality. 

A block flowchart can better describe the procedure taken to solve the equations

Author Response

Author response to Reviewer 4

Some figures need attention in terms of quality and proportionality.

We have made efforts to improve the quality and proportionality of the figures.

A block flowchart can better describe the procedure taken to solve the equations

The authors found this approach could not produce a more clear explanation of this particular derivation.